# 1-Bit FQT: Pushing the Limit of Fully Quantized Training to 1-bit

## Abstract

Fully quantized training (FQT) accelerates the training of deep neural networks by quantizing the activations, weights, and gradients into lower precision. To explore the ultimate limit of FQT (the lowest achievable precision), we make a first attempt to 1-bit FQT. We provide a theoretical analysis of FQT based on Adam and SGD, revealing that the gradient variance influences the convergence of FQT. Building on these theoretical results, we introduce an Average 1-bit Quantization (AQ) strategy. The strategy leverages the heterogeneity of gradients to mitigate gradient variance by pruning less informative gradients and enhancing the numerical precision of remaining gradients. Additionally, we propose Sample Channel joint Quantization (SCQ), which utilizes different quantization strategies in the computation of weight gradients and activation gradients to ensure that the method is friendly to low-bitwidth hardware. Finally, we present a framework to deploy our algorithm. For fine-tuning VGGNet-16 and ResNet-18 on multiple datasets, our algorithm achieves an average accuracy improvement of approximately 6%, compared to per-sample quantization. Moreover, our training speedup can reach a maximum of 5.13× compared to full precision training.

## 1 Introduction

Training neural networks has a high computational cost and memory footprint. Training with low-precision arithmetic (a.k.a., fully quantized training or FQT) can enhance computational and memory efficiency. FQT quantizes weights, activations, and gradients into low-bitwidth numerical formats, enabling a fast implementation of both forward and backward propagation on low-precision hardware.

The speedup potential of FQT depends on the numerical precision. Research aims to reduce the training numerical precision, without compromising convergence speed or accuracy. The required precision has been reduced from FP/INT16 (Micikevicius et al., 2017; Das et al., 2018) to FP/INT8 (Wang et al., 2018b; Banner et al., 2018; Zhu et al., 2020; Yang et al., 2020). As of now, some work (Sun et al., 2020; Chmiel et al., 2021; Xi et al., 2023) have successfully pushed precision down to 4 bits.

As the training numerical precision continues to decrease, a natural question arises:

*What is the ultimate limit of FQT (i.e., the minimum achievable bitwidth)?*

Answering this question not only advances our understanding of FQT but also provides a crucial direction for future hardware design strategies. Ideally, if we can push the bitwidth down to 1-bit, the training can be implemented with binary operations, such as XNOR and bitcounting operations (Courbariaux et al., 2016), and hardware design might be greatly simplified. Binary computation is already shown possible for *inference* acceleration, such as XNOR-Net (Rastegari et al., 2016), but 1-bit *training* remains unexplored.

Reducing the bitwidth for FQT is challenging because of (1) the lack of theoretical understanding, especially how gradient quantization affects the convergence; (2) the large quantization error of gradients, which causes a sharp performance drop or even divergence when reducing gradient bitwidth lower than 4-bit (Fig. 1). Due to these challenges, the current research frontier is still 4-bit FQT.

In this work, we make a first attempt towards achieving 1-bit FQT. Firstly, we provide a theoretical analysis for FQT based on both Adam (Kingma & Ba, 2014) and SGD. Our analysis links the convergence with gradient variance. Specifically, our analysis reveals that Adam is more suitable for FQT than SGD in the low-bitwidth regime, due to their different sensitivity to gradient variance.

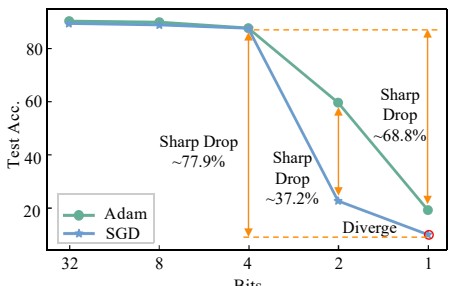

Figure 1: Gradient numerical precision ("bits") vs. test accuracy of VGGNet16 on CIFAR-10, trained with Adam and SGD. (**The supplementary results are in Fig. 8**)

Inspired by the above theory, we propose a hardware-friendly algorithm for 1-bit FQT. Our algorithm, composed of an Activation Gradient Pruning (AGP) and per-group quantization (Chen et al., 2020; Cho & Yoo, 2020), effectively reduces gradient variance. AGP utilizes gradient heterogeneity by discarding less informative groups and allocating saved resources to improve the numerical precision of more informative ones. Additionally, we propose Sample Channel joint Quantization (SCQ), an effective quantization scheme for accelerated performance. SCQ employs different quantization methods for computing weight gradients and activation gradients, ensuring both can be effectively implemented on low-bitwidth computing units.

We examine the potential of 1-bit FQT on transfer learning tasks in both vision and NLP domain. In this task, 1-bit FQT algorithm is used for on-device finetuning a pretrained 1-bit model to adapt new data. On all the datasets, our 1-bit FQT algorithm can successfully *converge* and demonstrate significantly superior performance compared to directly applying the previous FQT method to the task. The average accuracy drop on visual classification datasets is approximately 5%, compared to training the binary model with full-precision gradients. Notably, the average accuracy loss is negligible (less than 1%) on Flowers (Nilsback & Zisserman, 2008) dataset and Pets (Parkhi et al., 2012) dataset, indicating that 1-bit FQT might indeed be useful in some cases. We implement our algorithm on Hygon and Raspberry Pi devices as a PyTorch-based library `binop`. Accelerated on-device training can be achieved with simple layer substitution, e.g., replace `torch.nn.Conv2d` with `binop.Conv2d`. In practice, our method can achieve up to 5.13× speedup, compared to FP32 PyTorch. It is important to note that the primary aim of this paper is to explore the ultimate limit of Fully Quantized Training (FQT) rather than to focus on practical application performance. These results indicate that, in certain tasks, FQT precision can indeed be pushed to the extreme 1-bit level, offering valuable insights for future research.

## 2 RELATED WORKS

**Quantization Aware Training.** QAT is a method designed to accelerate *inference* by quantizing the activations and weights. Various works (Zhou et al., 2017; Choi et al., 2018; Zhang et al., 2018; Jacob et al., 2018; Dong et al., 2019; Tang et al., 2022; Liu et al., 2023) have been developed to quantize weights and activations into lower bitwidth. Furthermore, some studies (Rastegari et al., 2016; Bulat & Tzimiropoulos, 2019; Wang et al., 2020; Bai et al., 2020; Wu et al., 2023; Qin et al., 2023) have reduced the numerical precision of weights and activation values to 1 bit. However, QAT does not quantize gradients, and as a result, the backward propagation cannot be accelerated.

**Fully Quantized Training.** FQT further quantizes the gradients into lower precision, compared with QAT. Hence, FQT allows for efficient implementation of both forward and backward propagation on low-bitwidth computational units. FQT, unlike optimizer quantization (Lin et al., 2022a), involves quantizing weights, activations, and gradients altogether. Optimizer quantization only quantizes weight update (weight gradients), thus reducing communication costs but not accelerating computation (Saha et al., 2022). Early works on FQT use FP16 (Gupta et al., 2015; Micikevicius et al., 2017) or INT16 (Das et al., 2018) values to constrain weights, activations, and gradients. After that, various 8-bit numerical formats (Wang et al., 2018b; Banner et al., 2018; Zhu et al., 2020; Yang et al., 2020; Xi et al., 2024) have been proposed that further push the bitwidth of data to 8 bits. Subsequently, Chen et al. (2020) provides theoretical bounds on how the quantization scheme (bitwidth, type of quantizer) affects the quality of the quantized gradient. Based on that, some works have successfully

trained several networks with 4-bit activations/weights/gradients (Sun et al., 2020; Chmiel et al., 2021; Xi et al., 2023). The current research frontier is 4-bit FQT, but it still is not the ultimate limit.

## 3 FRAMEWORK

To better describe our approach, necessary notations are introduced first. We denote the DNN model composed of $L$ layers with the learnable parameter $\boldsymbol{\Theta}$ as $\mathbf{F}(.;\boldsymbol{\Theta})$. In each training iteration, we sample a minibatch $(\mathbf{X}, \mathbf{Y})$ from the dataset and input it into the model. The process is

$$\mathbf{H}^{(0)} = \mathbf{X}, \mathbf{H}^{(l)} = \mathbf{F}^{(l)}\left(\mathbf{H}^{(l-1)}; \boldsymbol{\Theta}^{(l)}\right), \forall l \in [L]_+, \tag{1}$$

where $\mathbf{H}^{(l)} \in \mathbb{R}^{N \times D^{(l)}}$ is a feature map ($N$ is the batch size, $D^{(l)}$ is the number of features), and $[L]_+ = \{1, 2, \ldots, L\}$ are sets of integers. $\mathbf{F}^{(l)}$ is the $l$-th layer of the model with parameter $\boldsymbol{\Theta}^{(l)}$. Given the minibatch loss $\mathcal{L}(\mathbf{H}^{(L)}, \mathbf{Y})$, we compute the gradient $\nabla_{\boldsymbol{\Theta}^{(l)}}\mathcal{L}$, and update the parameter. For simplicity, we use $\nabla_{\mathbf{H}^{(l)}}$ and $\nabla_{\boldsymbol{\Theta}^{(l)}}$ represent the activation/parameter gradient. The back-propagation is $\nabla_{\mathbf{H}^{(l-1)}}, \nabla_{\boldsymbol{\Theta}^{(l)}} = \mathbf{B}^{(l)}(\nabla_{\mathbf{H}^{(l)}}, \mathbf{H}^{(l-1)}, \boldsymbol{\Theta}^{(l)})$, where the function $\mathbf{B}^{(l)}(\cdot)$ takes the gradient of the output $\nabla_{\mathbf{H}^{(l)}}$ and the information kept in memory ($\mathbf{H}^{(l)}, \boldsymbol{\Theta}^{(l)}$), and computes the gradient of the input. For example, consider a linear layer $\mathbf{H}^{(l)} = \mathbf{H}^{(l-1)}\boldsymbol{\Theta}^{(l)}$ and its gradient is

$$\nabla_{\mathbf{H}^{(l-1)}} = \nabla_{\mathbf{H}^{(l)}}\boldsymbol{\Theta}^{(l)\top}, \quad \nabla_{\boldsymbol{\Theta}^{(l)}} = \mathbf{H}^{(l-1)\top}\nabla_{\mathbf{H}^{(l)}}. \tag{2}$$

### 3.1 QUANTIZED TRAINING

Here, we describe Quantization-Aware Training (QAT) and Fully Quantized Training (FQT). QAT is employed to accelerate *inference*, while FQT is designed to accelerate both inference and *training*.

Before embarking on QAT, the initial step involves quantizing the parameters and activations of the model:

$$\overline{\mathbf{H}}^{(l-1)} = Q_f(\mathbf{H}^{(l-1)}), \overline{\boldsymbol{\Theta}}^{(l)} = Q_{\boldsymbol{\Theta}}(\boldsymbol{\Theta}^{(l)}), \forall l \in [L]_+,$$

where $Q_f(\cdot)$ and $Q_{\boldsymbol{\Theta}}(\cdot)$ are quantizers for activations and weights, and $\overline{\mathbf{H}}^{(l-1)}$ and $\overline{\boldsymbol{\Theta}}^{(l)}$ are quantized activations and weights. The forward propagation Eq. 1 is quantized as $\forall l \in [L]_+, \mathbf{H}^{(l)} = \mathbf{F}^{(l)}(\overline{\mathbf{H}}^{(l-1)}; \overline{\boldsymbol{\Theta}}^{(l)})$, where $\overline{\mathbf{H}}^{(l-1)}$ and $\overline{\boldsymbol{\Theta}}^{(l)}$ represent low-bit data. Therefore, the inference can be efficiently implemented on low-bitwidth computing kernels. QAT leverages the straight-through estimator (Bengio et al., 2013) to train quantized models. The back-propagation Eq. 2 becomes:

$$\tilde{\nabla}_{\mathbf{H}^{(l-1)}} = \nabla_{\mathbf{H}^{(l)}}\overline{\boldsymbol{\Theta}}^{(l)\top}, \tilde{\nabla}_{\boldsymbol{\Theta}^{(l)}} = \overline{\mathbf{H}}^{(l-1)\top}\nabla_{\mathbf{H}^{(l)}}.$$

Since gradients are not quantized, the backpropagation cannot be accelerated.

The forward propagation of FQT is identical to QAT, FQT further quantizes the gradients at each layer. We use $\hat{\nabla}_{\mathbf{H}^{(l)}}$ and $\hat{\nabla}_{\boldsymbol{\Theta}^{(l)}}$ to represent the FQT gradient. The backpropagation is quantized as

$$\hat{\nabla}_{\mathbf{H}^{(l-1)}} = Q_g(\hat{\nabla}_{\mathbf{H}^{(l)}})\overline{\boldsymbol{\Theta}}^{(l)\top}, \hat{\nabla}_{\boldsymbol{\Theta}^{(l)}} = \overline{\mathbf{H}}^{(l-1)\top}Q_g(\hat{\nabla}_{\mathbf{H}^{(l)}}),$$

where $\hat{\nabla}_{\mathbf{H}^{(L)}} := \nabla_{\mathbf{H}^{(L)}}$, and $Q_g(\cdot)$ is a quantizer for gradients. Now, with all operands quantized, the backpropagation can be efficiently implemented on low-bitwidth kernels.

### 3.2 FQT WITH UNBIASED QUANTIZER

In our framework, $Q_f(\cdot)$ and $Q_{\boldsymbol{\Theta}}(\cdot)$ are deterministic quantizers, while $Q_g(\cdot)$ is an unbiased quantizer. This configuration follows Chen et al. (2020). In this framework, the gradients in FQT are unbiased estimates of QAT, ensuring both converge to the same point in expectation.

Consider $Q_g$ as an unbiased stochastic quantizer, i.e., $\mathbb{E}[Q_g(\nabla_{\mathbf{H}})] = \nabla_{\mathbf{H}}$, for any $\nabla_{\mathbf{H}}$, which are already widely adopted in existing FQT approaches (Banner et al., 2018; Xi et al., 2023), thereby enabling $\mathbb{E}[\hat{\nabla}_{\mathbf{H}^{(l)}}] = \nabla_{\mathbf{H}^{(l)}}$. The activation gradients of FQT is $\mathbb{E}[\hat{\nabla}_{\mathbf{H}^{(l-1)}}] = \mathbb{E}[\hat{\nabla}_{\mathbf{H}^{(l)}}]\overline{\boldsymbol{\Theta}}^{(l)\top} = \tilde{\nabla}_{\mathbf{H}^{(l-1)}}$, which implies FQT and QAT convergence to a stationary point in expectation. Given an

activation gradient tensor $\nabla_{\mathbf{H}}$, we quantize it to $b$-bit. We first compute the range of the tensor, and scale each element: $\overline{\nabla}_{\mathbf{H}_{i,j}} = \text{SR}(B(\nabla_{\mathbf{H}_{i,j}} - Z)/R)$, where $B = 2^b - 1$ are the number of quantization bins, $R = \max\{\nabla_{\mathbf{H}}\} - \min\{\nabla_{\mathbf{H}}\}$ is the range, $Z = \min\{\nabla_{\mathbf{H}}\}$ is the zero point, the stochastic rounding (Courbariaux et al., 2015) operation $\text{SR}(\cdot)$ convert input to integers, and $\overline{\nabla}_{\mathbf{H}_{i,j}}$ is the gradient quantized to $b$ bits. The dequantization is $\hat{\nabla}_{\mathbf{H}_{i,j}} = \overline{\nabla}_{\mathbf{H}_{i,j}} R/B + Z$. Due to the utilization of stochastic rounding, it is clear that $\mathbb{E}[\hat{\nabla}_{\mathbf{H}_{i,j}}] = \nabla_{\mathbf{H}_{i,j}}$.

The unbiased quantizer widely adopted in FQT is the per-group quantizer, including per-tensor quantizer (PTQ) (Banner et al., 2018), per-sample quantizer (PSQ) (Chen et al., 2020), and per-channel quantizer (PCQ) (Cho & Yoo, 2020). In these strategies, each group computes its own range and zero point, rather than sharing a common one, which addresses the large variation of dynamic range across groups.

## 4 THEORETICAL RESULTS

In this section, we analyze the convergence behavior of FQT under two different optimizers, Adam and SGD. The proof of theorems follows the framework in Kingma & Ba (2014), which can be found in Appendix A.

### 4.1 OPTIMIZER IMPACT ON CONVERGENCE

Quantized training with the Adam optimizer achieved much higher accuracy than those with SGD (Fig. 1). Although some prior studies (Bulat & Tzimiropoulos, 2019; Lin et al., 2022b) have highlighted this issue, the theoretical understanding of FQT with Adam is still lacking. To fill this gap, we will provide theoretical bounds on the convergence of FQT based on both Adam and SGD optimizers in the following part. (**The supplementary results are in Appendix B**))

We use the framework proposed in Zinkevich (2003) to analyze the convergence. We adopt the assumption made by Zinkevich (2003) that the loss function $\mathcal{L}$ is convex. At each iteration $t$, we predict using the parameter $\mathbf{\Theta}_t$ and evaluate it on the loss function $\mathcal{L}_t$. We evaluate the convergence of FQT using the regret: $R(T) = \sum_{t=1}^{T} [\mathcal{L}_t(\mathbf{\Theta}_t) - \mathcal{L}_t(\mathbf{\Theta}^*)]$, where $\mathbf{\Theta}^*$ are the best fixed point parameter. We define $\nabla_{\mathbf{\Theta}_{1:t,i}} \in \mathbb{R}^t$ as a vector that contains the $i$-th dimension of the gradients over all iterations till $t$, $\nabla_{\mathbf{\Theta}_{1:t,i}} = [\nabla_{\mathbf{\Theta}_{1,i}}, \nabla_{\mathbf{\Theta}_{2,i}}, \ldots, \nabla_{\mathbf{\Theta}_{t,i}}]$, $\hat{\nabla}_{\mathbf{\Theta}_{1:t,i}}$ is the quantized version of $\nabla_{\mathbf{\Theta}_{1:t,i}}$.

**Assumption 4.1** *There exists $\sigma, e > 0$, such that $\forall \mathbf{\Theta}_{t,i}$, $\text{Var}\left[\hat{\nabla}_{\mathbf{\Theta}_{t,i}}\right] \leq \sigma^2$, $-e \leq \mathbb{E}\left[\hat{\nabla}_{\mathbf{\Theta}_{t,i}}\right] \leq e$.*

**Assumption 4.2** *The distance between any $\mathbf{\Theta}_t$ is bounded, $\|\mathbf{\Theta}_n - \mathbf{\Theta}_m\|_2 \leq D$, $\|\mathbf{\Theta}_n - \mathbf{\Theta}_m\|_\infty \leq D_\infty$, for any $m, n \in \{1, \ldots, T\}$.*

Given an unbiased gradient, we now establish the convergence of quantized training under SGD. The iteration form of SGD is $\mathbf{\Theta}_{t+1} \leftarrow \mathbf{\Theta}_t - \alpha_t \hat{\nabla}_{\mathbf{\Theta}_t}$.

**Theorem 4.3** *If Assumption 4.1 and 4.2 holds, let $\alpha_t = \frac{\alpha}{\sqrt{t}}$ and the number of elements in the gradient is $d$. SGD achieves the following guarantee, for all $T \geq 1$. $R^{SGD}(T) \leq \frac{D^2}{2\alpha} + \frac{\alpha T d(\sigma^2 + e^2)}{2}$.*

The iteration form of Adam is expressed as follows:

$$\begin{cases} m_t = \beta_{1,t} \cdot m_{t-1} + (1 - \beta_{1,t}) \cdot \hat{\nabla}_{\mathbf{\Theta}_t}, v_t = \beta_2 \cdot v_{t-1} + (1 - \beta_2) \cdot \left(\hat{\nabla}_{\mathbf{\Theta}_t}\right)^2, \\ \hat{m}_t = \frac{m_t}{1 - \beta_1^t}, \hat{v}_t = \frac{v_t}{1 - \beta_2^t}, \mathbf{\Theta}_{t+1} = \mathbf{\Theta}_t - \frac{\alpha}{\sqrt{\hat{v}} + \epsilon} \cdot \hat{m}_t. \end{cases}$$

**Assumption 4.4** *The function $\mathcal{L}_t$ has bounded gradients, $\forall \mathbf{\Theta}$, $\left\|\hat{\nabla}_{\mathbf{\Theta}_t}\right\|_2 \leq G$, $\left\|\hat{\nabla}_{\mathbf{\Theta}_t}\right\|_\infty \leq G_\infty$.*

**Theorem 4.5** *If Assumption 4.1, 4.2 and 4.4 holds, let $\beta_1, \beta_2 \in [0,1)$ satisfy $\frac{\beta_1^2}{\sqrt{\beta_2}} < 1$, $\alpha_t = \frac{\alpha}{\sqrt{t}}$, and $\beta_{1,t} = \beta_1 \lambda^{t-1}, \lambda \in (0,1)$. Adam achieves the following guarantee, for all $T \geq 1$.*
$$R^{Adam}(T) \leq \frac{((1-\lambda)^2 D^2 T + D_\infty^2) d}{2\alpha(1-\beta_1)(1-\lambda)^2} \sqrt{\sigma^2 + e^2} + \frac{\alpha(1+\beta_1) G_\infty \sqrt{T} d}{(1-\beta_1)\sqrt{1-\beta_2}(1-\gamma)^2} \sqrt{\sigma^2 + e^2}.$$

Based on Theorem 4.3, 4.5, Adam and SGD achieve the following guarantee, for $T \to \infty$.

$$\frac{R^{SGD}(T)}{T} \leq \alpha d(\sigma^2 + e^2)/2, \frac{R^{Adam}(T)}{T} \leq \frac{D^2 d}{2\alpha (1 - \beta_1)} \sqrt{\sigma^2 + e^2}.$$

From the inquation, it is straightforward to conclude that $\frac{R^{SGD}(T)}{T} = O(\sigma^2) + O(1)$, $\frac{R^{Adam}(T)}{T} = O(\sigma) + O(1)$. This implies that the convergence of FQT based on both Adam and SGD is influenced by the gradient variance, with SGD being more sensitive to variations in gradient variance.

### 4.2 QUANTIZER IMPACT ON GRADIENT VARIANCE

Based on our theory, gradient variance plays a crucial role in convergence. Gradient variance is primarily composed of two components: the variance of QAT gradients and the variance introduced by the gradient quantizers. Chen et al. (2020) reduced the complicated problem of gradient variance into the simple problem of quantizer variance. Thus, we need to minimize the quantizer variance.

The fundamental form of an unbiased quantizer $Q_g$ is given by Sec. 3.2, and its variance is $\text{Var}[Q_g(\hat{\nabla}_{\mathbf{H}^{(l)}}) \mid \hat{\nabla}_{\mathbf{H}^{(l)}}] = \frac{R^2}{B^2} \text{Var}[\text{SR}(\cdot) \mid \hat{\nabla}_{\mathbf{H}^{(l)}}] \leq \frac{N D^{(l)}}{4 B^2} R^2$, where the maximum variance of stochastic rounding $\text{SR}(\cdot)$ is 1/4. The expression reveals that as the bitwidth $b$ decreases, the variance significantly increases. Furthermore, due to the sensitivity of SGD to gradient variance, SGD performs less effectively than Adam in low precision scenarios (large gradient variance) (Fig. 1). Therefore, in scenarios with larger gradient variances, such as in quantized training, the Adam optimizer is recommended. Additionally, the variance is highly sensitive to the gradient range $R$, with outliers in the gradient expanding the range and consequently increasing the quantizer's variance.

## 5 1-BIT FQT ALGORITHM

In this section, we propose our 1-bit FQT algorithm, including the quantization of weights, activation, and gradients.

### 5.1 FORWARD PROPAGATION

In the forward propagation, both $Q_f$ and $Q_\Theta$ are deterministic quantizers, taking the form: $\text{sign}(x) = -1$ if $x \leq 0$ otherwise 1. For a fully connected layer, the forward propagation is $\mathbf{H}^{(l)} = (\text{sign}(\mathbf{H}^{(l-1)}) \text{sign}(\mathbf{\Theta}^{(l)})) \odot \Gamma$, where $\Gamma \in \mathbb{R}^{D^{(l)}}$ represents the shared scaling factor for both weights and activations, and it is learnable parameters. The form follows Bulat & Tzimiropoulos (2019).

### 5.2 BACKWARD PROPAGATION

The form of backpropagation is

$$\hat{\nabla}_{\mathbf{H}^{(l-1)}} = Q_g(\hat{\nabla}_{\mathbf{H}^{(l)}}) \text{sign}(\mathbf{\Theta}^{(l)^\top}), \hat{\nabla}_{\mathbf{\Theta}^{(l)}} = \text{sign}(\mathbf{H}^{(l-1)^\top}) Q_g(\hat{\nabla}_{\mathbf{H}^{(l)}}). \tag{3}$$

Based on our theory, reducing quantizer variance is crucial to ensure the convergence of the model. However, outliers in the gradients can widen the range of gradients, thereby increasing variance.

To mitigate the impact of outliers on variance, per-group quantization is widely employed. Per-group quantization reduces variance by assigning a separate range to each group instead of sharing a large range among all. For example, we perform per-sample quantization on $\hat{\nabla}_{\mathbf{H}^{(l)}} \in \mathbb{R}^{N \times D^{(l)}}$ and its form is $Q_g(\hat{\nabla}_{\mathbf{H}^{(l)}}) = \mathbf{S}^{(l)}(\text{SR}((\mathbf{S}^{(l)})^{-1}(\hat{\nabla}_{\mathbf{H}^{(l)}} - \mathbf{Z}))) + \mathbf{Z}$, where $\mathbf{S}^{(l)} = \text{diag}\{R_1/B, ..., R_N/B\}$, $R_i, \mathbf{Z}_i$ represent the range and zero point of activation gradients for the $i$-th sample. Its variance is

$$\text{Var}[Q_g(\hat{\nabla}_{\mathbf{H}^{(l)}}) \mid \hat{\nabla}_{\mathbf{H}^{(l)}}] \leq \frac{D^{(l)}}{4 B^2} \sum_{i=1}^{N} R_i^2. \tag{4}$$

However, the variance of PSQ is still too large for 1-bit FQT.

To address this, we propose Average 1-bit Quantization (AQ), which consists of Activation Gradient Pruning (AGP) and Per-group Quantization, to reduce quantizer variance by utilizing the heterogeneity in gradient distributions (Xi et al., 2023). Gradients exhibit varying ranges across samples, with some having large ranges and others much smaller, a pattern that also holds across the channel dimension, as illustrated in Fig. 2. Groups (samples or channels) with smaller gradient ranges tend to have values close to zero, indi-

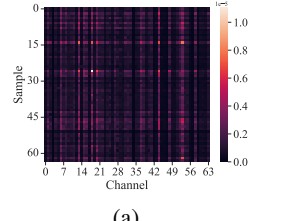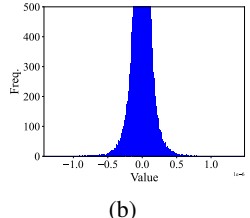

(a)                                    (b)

Figure 2: Heterogeneity in a ResNet18's gradients. (a) Heatmap of the per-group range at the conv2.1.2 layer; (b) Histogram of the gradient in a certain group.

cating that less information stored in these groups. By pruning these less informative groups, we can reallocate the saved computational resources to groups with larger ranges (increased bitwidth). As shown by Eq. 4, variance primarily originates from groups with larger ranges ($R$), and it is highly sensitive to numerical precision. Therefore, by using higher numerical precision (i.e., increased bitwidth) for these groups, we can effectively reduce the overall variance.

Achieving AQ based on the above idea requires ensuring three conditions: (1) if the bitwidth of retained groups is $b$, only $1/b$ of the groups can be preserved, thereby maintaining an average bitwidth of 1; (2) adopting random pruning to ensure the unbiased nature of quantization; (3) groups with larger ranges are more likely to be retained. Based on that, we first assign each group a probability $p_i \in [0, 1], i = 1, \cdots, N$. To retain $\frac{N}{b}$ groups and ensure the retained groups have a large range, $p_i$ needs to satisfy $\sum_{i=1}^{N} p_i = \frac{N}{b}$ and $p_i \propto R_i$, i.e., $p_i = \frac{NR_i}{bR_{total}}$, $R_{total} = \sum_{i=1}^{N} R_i$. Then we define random masks $m_i \sim \text{Bern}(p_i)$ to prune unimportant groups, and perform per-group quantization on the remaining ones. Its form is: $Q_g(\hat{\nabla}_{\mathbf{H}^{(l)}}) = Q_{PSQ}^b(\mathbf{M}\hat{\nabla}_{\mathbf{H}^{(l)}})$, where $\mathbf{M} = \text{diag}(\frac{m_1}{p_1}, \ldots, \frac{m_N}{p_N})$, $Q_{PSQ}^b$ is $b$-bit PSQ. $Q_g$ is an unbiased quantizer since $\mathbb{E}[Q_{PSQ}^b(\mathbf{M}\hat{\nabla}_{\mathbf{H}^{(l)}})] = \mathbb{E}[\mathbf{M}]\hat{\nabla}_{\mathbf{H}^{(l)}} = \mathbf{I}\hat{\nabla}_{\mathbf{H}^{(l)}}$. The variance is

$$\text{Var}\left[Q_g\left(\hat{\nabla}_{\mathbf{H}^{(l)}}\right) \mid \hat{\nabla}_{\mathbf{H}^{(l)}}\right] \leq \frac{D^{(l)}}{4B^2} \sum_{i=1}^{\frac{N}{b}} R_i^2. \tag{5}$$

From Eq. 5, it can be observed that the variance of AQ is significantly smaller than that of 1-bit PSQ ($\frac{D^{(l)}}{4B^2} \sum_{i=1}^{N} R_i^2$). The proof is given in Appendix C.

Despite reducing the average precision of the gradient to 1 bit, the binarized operations are limited because the retained groups remain non-binarized ($b$-bit). We perform a splitting operation to transform the gradient into a format suitable for binarized operations. For example, a value of 2 (binary: 10) in a 2-bit tensor $\overline{\nabla}_{\mathbf{H}^{(l)}}$ is split into 1 in $\overline{\nabla}_{\mathbf{H}^{(l)}}^{\uparrow}$ and 0 in $\overline{\nabla}_{\mathbf{H}^{(l)}}^{\downarrow}$, $\overline{\nabla}_{\mathbf{H}^{(l)}} = \overline{\nabla}_{\mathbf{H}^{(l)}}^{\uparrow} \times 2 + \overline{\nabla}_{\mathbf{H}^{(l)}}^{\downarrow}$. The Eq. 3 can be rewritten as:

$$\hat{\nabla}_{\mathbf{H}^{(l-1)}} = (\mathbf{S}^{(l)}(\overline{\nabla}_{\mathbf{H}^{(l)}}^{\uparrow} \times 2 + \overline{\nabla}_{\mathbf{H}^{(l)}}^{\downarrow}) + \mathbf{Z})(\overline{\mathbf{\Theta}}^{(l)^{\top}}), \hat{\nabla}_{\mathbf{\Theta}^{(l)}} = (\overline{\mathbf{H}}^{(l-1)^{\top}})(\mathbf{S}^{(l)}(\overline{\nabla}_{\mathbf{H}^{(l)}}^{\uparrow} \times 2 + \overline{\nabla}_{\mathbf{H}^{(l)}}^{\downarrow}) + \mathbf{Z}), \tag{6}$$

where $\overline{\mathbf{\Theta}}^{(l)^{\top}}$ and $\overline{\mathbf{H}}^{(l-1)^{\top}}$ represent binary weight and activation. Due to the removal of some groups, the shape of the result differs from the original, and we fill the gaps with zeros. The format conversion operation from {0,1} to {-1,1} is omitted here. The entire process is illustrated in Fig. 3.

## 5.3 PRACTICAL ACCELERATION

To ensure the compatibility of binary matrix multiplication (BMM) with low-bit hardware, we require that all tensors involved in matrix multiplication are binarized. From Eq. 6, it is evident that the computation of activation gradients $\hat{\nabla}_{\mathbf{H}^{(l-1)}}$ can be accelerated, as $\overline{\nabla}_{\mathbf{H}^{(l)}}^{\uparrow}(\overline{\mathbf{\Theta}}^{(l)^{\top}})$ and $\overline{\nabla}_{\mathbf{H}^{(l)}}^{\downarrow}(\overline{\mathbf{\Theta}}^{(l)^{\top}})$ can be efficiently implemented in hardware, whereas weight gradients $\hat{\nabla}_{\mathbf{\Theta}^{(l)}}$ cannot be accelerated due to the presence of floating-point tensors $\mathbf{S}^{(l)}$ in $\overline{\mathbf{H}}^{(l-1)^{\top}}(\mathbf{S}^{(l)})\overline{\nabla}_{\mathbf{H}^{(l)}}^{\uparrow}$ and $\overline{\mathbf{H}}^{(l-1)^{\top}}(\mathbf{S}^{(l)})\overline{\nabla}_{\mathbf{H}^{(l)}}^{\downarrow}$, making hardware implementation infeasible.

To address this issue, we propose Sample Channel joint Quantization (SCQ), wherein PCQ is employed during the computation of weight gradients, while PSQ is utilized for the computation of

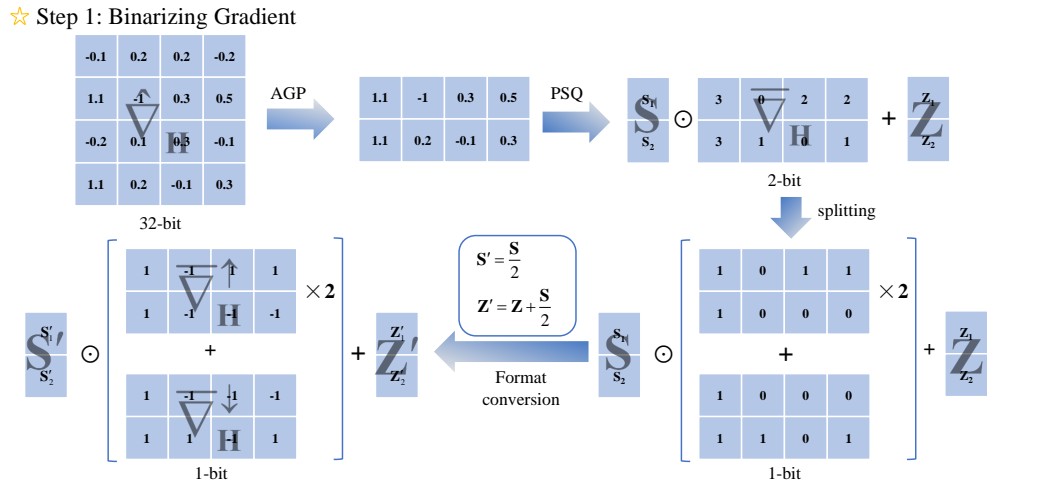

☆ Step 1: Binarizing Gradient

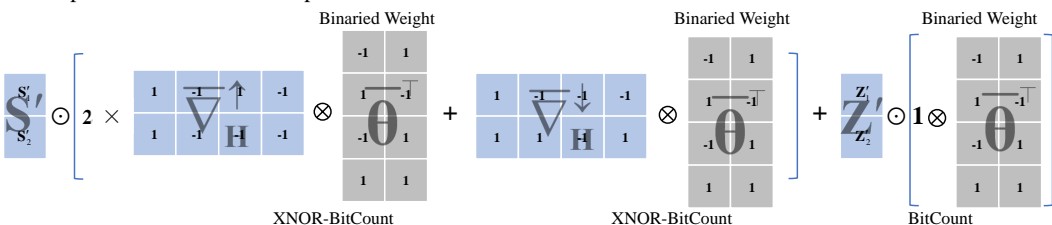

☆ Step 2: Backward with bit operations

Figure 3: The process of AQ and binary matrix multiplication. Here, we removed half of the groups, thus the bitwidth of the remaining groups is 2.

activation gradients. Building upon this quantization strategy, the computation of weight gradients can be rewritten as:

$$\hat{\nabla}_{\mathbf{\Theta}^{(l)}} = (\overline{\mathbf{H}}^{(l-1)^\top})((\overline{\nabla}^\uparrow_{\mathbf{H}^{(l)}_{PCQ}} \times 2 + \overline{\nabla}^\downarrow_{\mathbf{H}^{(l)}_{PCQ}})\mathbf{S}^{(l)}_{PCQ} + \mathbf{Z}_{PCQ}),$$

where $\mathbf{S}^l_{PCQ} = \mathrm{diag}\left\{\frac{R^c_1}{B}, \dots, \frac{R^c_{D^{(l)}/2}}{B}\right\}$, $R^c_i$ represents the range of $i$-th channel. PCQ apply different scale and zero point per each channle of the gradient. This strategy facilitates the acceleration of both weight and activation gradient computations. The final formulation is:

$$\hat{\nabla}_{\mathbf{H}^{(l-1)}} = Q^b_{PSQ}\left(\mathbf{M}\hat{\nabla}_{\mathbf{H}^{(l)}}\right)(\overline{\mathbf{\Theta}}^{(l)^\top}), \hat{\nabla}_{\mathbf{\Theta}^{(l)}} = (\overline{\mathbf{H}}^{(l-1)^\top})Q^b_{PCQ}\left(\hat{\nabla}_{\mathbf{H}^{(l)}}\mathbf{M}_{PCQ}\right).$$

Since PCQ treats a channel as a group, pruning operations also need to be performed along the channel dimension. Due to space constraints, implementation details are provided in Appendix D.

# 6 EXPERIMENTS

We evaluate our approach on transfer learning tasks. Although our approach is constrained to transfer learning, it still holds practical value in on-device training (Lin et al., 2022b). Due to challenges such as environmental constraints and limited memory, it is impractical to perform training from scratch on edge devices (Ren et al., 2021). The experiment details and results from training from scratch are in Appendix E.

## 6.1 MAIN RESULTS

We employed two DNN architectures, ResNet18 (He et al., 2016) and VGGNet16 (Simonyan & Zisserman, 2014). We pre-trained them on ImageNet (Deng et al., 2009) and subsequently conducted QAT. The quantized models are fine-tuned on downstream datasets to evaluate our approach. Following Lin et al. (2022b), we utilize various datasets, including Cars (Krause et al., 2013), CIFAR-10 (Krizhevsky et al., 2009), CIFAR-100 (Krizhevsky et al., 2009), CUB (Welinder et al., 2010), Flowers (Nilsback & Zisserman, 2008) and Pets (Parkhi et al., 2012).

Table 1: Experimental results on multiple downstream datasets. "(W, A, G)" denote the bitwidth of weight, activations, and gradients, respectively. $b$ represents the bitwidth of the remaining groups.

| Method | Precision | Accuracy(%) | | | | | | |
|---|---|---|---|---|---|---|---|---|
| | (W, A, G) | CIFAR-10 | CIFAR-100 | Flowers | Cars | Pets | CUB | Average |
| | | | ResNet-18 | | | | | |
| QAT | 1, 1, 32 | 87.31±.25 | 65.82±.43 | 78.85±.80 | 50.81±.38 | 71.68±.21 | 42.13±.43 | 66.10 |
| PSQ | 1, 1, 1 | 71.04±.61 | 47.71±.98 | 78.91±.10 | 23.14±.91 | 68.93±.39 | 34.29±.62 | 54.01 |
| Ours ($b=2$) | 1, 1, 1 | 74.10±.21 | 52.19±.62 | **79.93**±.20 | 26.51±.76 | 70.47±.52 | 36.59±.31 | 56.63 |
| Ours ($b=4$) | 1, 1, 1 | **78.52**±.56 | **56.83**±.61 | 79.28±.50 | **37.88**±.36 | **71.17**±.16 | **39.47**±.25 | **60.53** |
| Ours ($b=8$) | 1, 1, 1 | 73.73±.99 | 52.64±.36 | 78.10±.65 | 29.78±.89 | 69.98±.32 | 37.01±.53 | 56.87 |
| | | | VGGNet-16 | | | | | |
| QAT | 1, 1, 32 | 89.80±.36 | 71.70±.17 | 86.86±.35 | 67.65±.03 | 79.49±.44 | 53.39±.57 | 74.82 |
| PSQ | 1, 1, 1 | 80.60±.20 | 59.81±.20 | 84.65±.05 | 40.01±.88 | 77.20±.38 | 43.17±.44 | 64.24 |
| Ours ($b=2$) | 1, 1, 1 | 82.66±.44 | 62.04±.01 | 85.75±.29 | 44.40±.92 | 77.77±.35 | 46.33±.53 | 66.49 |
| Ours ($b=4$) | 1, 1, 1 | **84.38**±.12 | **63.65**±.19 | **87.12**±.20 | **57.06**±.60 | **78.48**±.21 | **49.10**±.17 | **69.97** |
| Ours ($b=8$) | 1, 1, 1 | 78.14±.86 | 60.20±.08 | 86.24±.15 | 46.95±.21 | 77.39±.26 | 47.48±.20 | 66.07 |

Table 2: Experimental results under different numerical precisions.

| Method | Precision | Accuracy(%) | | | | | | |
|---|---|---|---|---|---|---|---|---|
| | (W, A, G) | CIFAR-10 | CIFAR-100 | Flowers | Cars | Pets | CUB | Average |
| QAT | 1, 1, 32 | 87.31 | 65.82 | 78.85 | 50.81 | 71.68 | 42.13 | 66.10 |
| PSQ | 1, 1, 2 | 74.55 | 53.30 | 79.39 | 30.02 | 70.92 | 36.94 | 57.52 |
| Ours | | **85.52** | **62.77** | **79.43** | **46.91** | **72.01** | **41.46** | **64.52** |
| PSQ | 1, 1, 4 | 86.86 | 65.35 | 79.21 | 49.65 | 71.85 | 42.06 | 65.83 |
| Ours | | **86.90** | **65.45** | **79.24** | **50.64** | **72.20** | **43.11** | **66.26** |

Table 3: Experimental results of the advanced binary model (Adabin (Tu et al., 2022)).

| Method | Accuracy(%) | | | | | | |
|---|---|---|---|---|---|---|---|
| | CIFAR-10 | CIFAR-100 | Flowers | Cars | Pets | CUB | Average |
| QAT | 92.17 | 71.81 | 92.35 | 74.54 | 83.02 | 57.35 | 78.54 |
| PSQ | 83.94 | 64.19 | 90.98 | 47.61 | 80.81 | 48.72 | 69.38 |
| Ours | **90.50** | **71.13** | **91.87** | **69.20** | **82.37** | **54.45** | **76.59** |

**Converged model accuracy.** To evaluate the performance of our method, we report the accuracy of two model architectures, VGG16 and ResNet18, across various datasets in Table 1. We report the mean and stddev of 3 runs. The compared approaches include QAT (Bulat & Tzimiropoulos, 2019) and PSQ. Since QAT employs training with full precision gradients, it can be considered as an upper bound for the accuracy of 1-bit FQT. Existing work has not tried 1-bit FQT, so we did not compare more methods. On VGGNet16, our method achieves $< 10\%$ average accuracy degradation across all configurations, as compared to the baseline QAT with 32-bit gradients. Moreover, in the optimal configuration ($b$=4), our method exhibits only approximately 5% average accuracy drop. On the more challenging ResNet18, the worst configuration ($b$=2) and the optimal configuration ($b$=4) achieves 9.47% and 5.57% average accuracy degradation, respectively, compared to QAT. Furthermore, on some datasets such as Flowers and Pets, our method exhibits minimal accuracy loss, indicating its suitability for these datasets. In summary, while our approach exhibits a notable decrease in accuracy compared to QAT, the incurred gap remains acceptable considering the benefits gained from reducing the numerical precision of gradients to 1 bit. Additionally, we compared our method with 1-bit PSQ. Across both frameworks, our approach consistently outperformed it in terms of average accuracy across all configurations. Moreover, except for individual outcomes in the worst configuration, our method also exhibited superior accuracy across all datasets.

**The value of $b$.** We investigate the impact of hyperparameter $b$ on performance and determine the optimal choice for $b$. From Eq. 5, as $b$ increases, the variance of the quantizer gradually decreases, suggesting an improvement in training convergence. However, the increase in $b$ also implies more discarded groups, leading to larger losses. Therefore, the choice of $b$ becomes a trade-off issue. In Table 1, we report the accuracy of our method across various datasets under three different configurations ($b = 2$, $b = 4$, and $b = 8$). On VGGNet16 and ResNet18, the configuration with $b = 4$ consistently outperforms the others ($b = 2$ and $b = 8$) in terms of average accuracy. Moreover, this observation extends to the majority of datasets, where, even on a few datasets, the results for the configuration with $b = 4$ may not be optimal, the performance difference remains marginal compared to the optimal results. In conclusion, the optimal configuration is $b = 4$.

**Generalizability.** To evaluate the generalization ability of our method, we conducted a series of experiments under various conditions. Table 2 presents the results under various precision settings (W, A, G). As observed from Table 2, the performance of both our method and PSQ improves significantly with increased numerical precision. Notably, our method surpasses PSQ in several datasets at higher precision settings. In addition, when the precision is set to 4 bits, the fully quantized training methods (PSQ and Ours) achieve similar performance to QAT. Therefore, 4-bit FQT can meet the requirements

Table 4: Training speedup of 1-bit FQT across different input resolutions. "Non-Full vs. Full" represents the speedup between non-full optimized (matrix partitioning only) 1-bit FQT and fully optimized FP32 training (PyTorch32). "Unoptimized vs. Unoptimized" shows the speedup between unoptimized 1-bit FQT and unoptimized FP32 training.

| Optimization Level | Model | Hygon | | | | | Raspberry Pi 5 | | |
|---|---|---|---|---|---|---|---|---|---|
| | | 32 | 64 | 128 | 224 | average | 32 | 64 | average |
| Non-Full vs. Full | VGGNet-16 | 5.13× | 3.71× | 3.38× | 2.73× | 3.74× | 3.72× | 1.25× | 2.49× |
| | ResNet-18 | 2.93× | 2.88× | 2.62× | 2.15× | 2.65× | 1.42× | 0.97× | 1.20× |
| Unoptimized vs. Unoptimized | VGGNet-16 | 109.0× | 108.3× | 106.3× | 93.0× | 104.2× | 69.4× | 58.5× | 64.0× |
| | ResNet-18 | 89.7× | 85.9× | 77.0× | 65.0× | 79.4× | 66.5× | 57.5× | 62.0× |

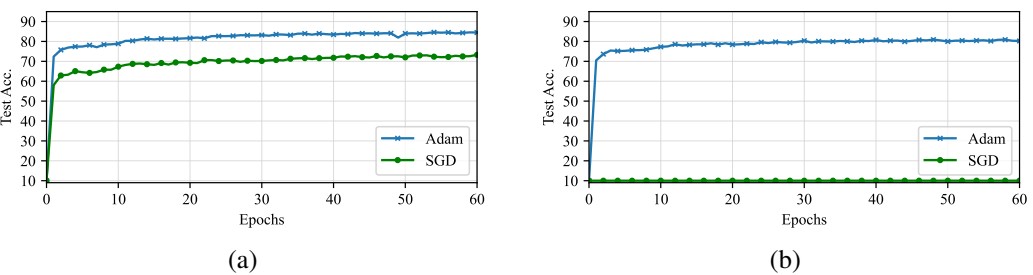

(a)                  (b)

Figure 4: Our method (a) vs. PSQ (b): Testing accuracy comparison on VGGNet16 for CIFAR-10. (**The supplementary results are shown in Fig. 9.**)

of performance-critical applications, where both computational efficiency and model accuracy are essential. Table 3 shows the results of binary training based on a more advanced binary model (Adabin (Tu et al., 2022)). As seen from Table 3, our method consistently outperforms PSQ across multiple datasets in the context of binary training with this more advanced model. Furthermore, compared to training with XNOR-Net, the performance gap between our method and QAT is significantly reduced, indicating that ours maintains strong generalization even when training advanced binarized models.

**Effect of the optimizer.** To validate our theory that the SGD optimizer is more sensitive to the variance of gradients compared to the Adam optimizer, we conduct a performance comparison of different optimizers on the CIFAR-10 dataset. We present the test accuracy curves of our method and PSQ across different optimizers in Fig. 4. For both methods, model performance degrades when using the SGD compared to the Adam. This is primarily attributed to the sensitivity of SGD to gradient variance. In addition, we observed that our method with SGD experienced only a modest accuracy drop, whereas the PSQ method with SGD failed to converge entirely. We attribute this observation to the larger variance introduced by PSQ compared to our quantizer, resulting in divergence.

**Variance.** To demonstrate the advantages of our quantizer in reducing variance, we present the quantizer variance of ResNet18 in Fig. 5. In general, the quantizer variance of our method is lower than that of PSQ across all datasets. Additionally, the variance on the Flowers and Pets is lowest, explaining why the impact of quantization on accuracy is minor for them.

**Training from scratch.** In this experiment, we applied more aggressive settings to explore the feasibility of 1-bit FQT in challenging scenarios, such as training from scratch and on large-scale datasets like ImageNet. We trained two binary models from scratch in Table 9, XNOR-Net++ and Adabin. The results show that while our method consistently outperforms PSQ across multiple datasets, there remains a significant perfor-

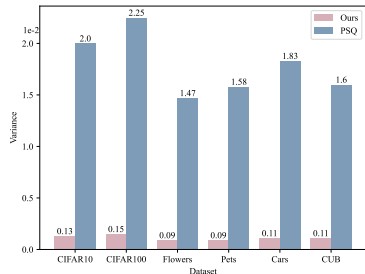

Figure 5: Quantizer variances across different datasets.

mance gap between QAT and our 1-bit FQT method. Therefore, 1-bit FQT is only feasible for transfer learning and still faces significant challenges in training from scratch. We analyzed the reasons for the gap between training from scratch and fine-tuning, and found that in the former scenario, the gradient range is significantly larger, leading to increased variance and greater difficulty in convergence, as shown in Fig. 7.

Table 5: Object detection on PASCAL VOC, classification on CIFAR-100 and NLP tasks on GLUE.

| Task | Model | Method | Bits | mAP/Acc./Avg. |
|---|---|---|---|---|
| Det. | Faster R-CNN | QAT | 32 | 52.34 |
| | Faster R-CNN | Ours | 1 | 50.68 |
| Cls. | MLP-Mixer | QAT | 32 | 52.17 |
| | MLP-Mixer | Ours | 1 | 48.65 |
| NLP | BERT | QAT | 32 | 63.20 |
| | BERT | Ours | 1 | 54.81 |

Table 6: Average 1-bit vs. 1-bit. The running time of matrix multiplication involving various sizes.

| Setting | Time(ms) across various sizes | | | | | |
|---|---|---|---|---|---|---|
| | 512 512 512 | 512 512 1024 | 1024 512 512 | 1024 512 1024 | 2048 512 512 | 2048 512 1024 |
| Average 1-bit | 8.40 | 15.92 | 16.61 | 31.03 | 32.22 | 61.33 |
| 1-bit | 8.01 | 14.54 | 15.91 | 29.79 | 29.92 | 59.94 |

**Other results.** We report results for other architectures and tasks in Table 5. The details can be found in Appendix E. On Faster R-CNN (Ren et al., 2015), our approach with 1-bit gradients achieves 1.66% mAP degradation, as compared to the baseline QAT with 32-bit gradients. In addition, for Mixer-MLP (Tolstikhin et al., 2021), an all-MLP architecture, our approach shows a decrease of 3.52% in classification accuracy compared to the baseline. For BERT, our approach achieves 8.39% average performance degradation. These results indicate the potential of our approach to transfer to other architectures and tasks. We did not extend the 1-bit FQT to large models primarily because existing binarized networks (Huang et al., 2024) only quantize weights to 1 bit, while activations remain at higher precision, hindering hardware acceleration during training.

## 6.2 COMPUTATIONAL EFFICIENCY

We discuss the computational overhead of our method. Our implementation is not fully optimized, as the comprehensive hardware-algorithm co-design is beyond the scope of this paper. Our experiments are conducted on a single-core Hygon CPU and edge device (Raspberry Pi 5).

**Training speedup.** We compare the training time of the FP32 PyTorch and our 1-bit FQT for VGGNet16 and ResNet18. We vary the resolution of the input and summarize the speedup of our method in Table 4. For VGGNet16, our algorithm achieves an average speedup of 3.74× and 2.49× on the Hygon and edge device, respectively. For ResNet18, our algorithm achieves 2.65× and 1.20× average speedup. Additionally, to assess the acceleration potential of 1-bit FQT, we compare their speedup at the same optimization level (unoptimized). The results indicate that across multiple cases, the speedup is above a *hundredfold*. On edge devices, our method achieves a speedup of over 50×. This gap indicates significant acceleration potential for 1-bit FQT. Finally, we analyzed why the speedup of ResNet18 is lower than that of VGG16, concluding from Fig. 10 in Appendix E that our implementation is more favorable for layers with more filters, which leads to a higher speedup for VGG16, as it has a higher average number of filters per layer.

**Average 1-bit vs. 1-bit.** We compared the runtime of average 1-bit matrix multiplication and 1-bit matrix multiplication across different matrix sizes in Table 6. The results demonstrate that the difference in runtime between these two methods is minimal, indicating similarity in the runtime of our average 1-bit FQT and 1-bit FQT. The computational complexity analysis is provided in the Appendix D.

## 7 CONCLUSION

We propose a hardware-friendly 1-bit FQT method in this work, which pushes the limit of FQT. Through convergence analysis, we propose AGP to reduce the variance of the quantizer, thereby enhancing the convergence of quantized training. Subsequently, to address the issue of unacceleratable weight gradient computation, we present a SCQ strategy. Finally, we propose a framework that practically accelerates training, achieving a speedup of up to 5.13× compared to full precision training. While our approach focuses solely on convolutional neural networks in this study, experiments indicate its potential applicability to other architectures.

**Limitations:** The primary limitation of this work lies in its ability to achieve 1-bit FQT in transfer learning tasks but not in training from scratch. To the best of our knowledge, even the 3-bit FQT from scratch is still an open problem.

## REPRODUCIBILITY STATEMENT

All code used in our experiments is included in the supplementary materials to facilitate reproducibility. The theoretical results, along with detailed proofs and the analysis of assumptions used throughout the paper, are provided in Appendix A. Further implementation details, including hyperparameters and experimental configurations, can be found in Appendix E. By providing these resources, we aim to ensure that our findings can be easily reproduced and built upon by the research community.

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

## A  Proof of Theorems

**Lemma A.1** *If a function $\mathcal{L} : R^d \to R$ is convex, then for all $x, y \in R^d$,*

$$\mathcal{L}(y) \geq \mathcal{L}(x) + \nabla\mathcal{L}(x)^T (y - x).$$

**Lemma A.2** *Let $\hat{\nabla}_{\Theta_t} = \hat{\nabla}\mathcal{L}_t(\Theta_t)$ and $\hat{\nabla}_{\Theta_{1:t}}$ be defined as above and bounded, $\left\|\hat{\nabla}_{\Theta_t}\right\|_2 \leq G, \left\|\hat{\nabla}_{\Theta_t}\right\|_\infty \leq G_\infty$. Then,*

$$\sum_{t=1}^{T} \sqrt{\frac{\hat{\nabla}_{\Theta_{t,i}}^2}{t}} \leq 2G_\infty \left\|\hat{\nabla}_{\Theta_{1:T,i}}\right\|_2.$$

**Lemma A.3** *Let $\gamma \triangleq \frac{\beta_1^2}{\sqrt{\beta_2}}$. For $\beta_1, \beta_2 \in [0, 1)$ that satisfy $\frac{\beta_1^2}{\sqrt{\beta_2}} < 1$ and bounded $\hat{\nabla}_{\Theta_t}, \left\|\hat{\nabla}_{\Theta_t}\right\|_2 \leq G, \left\|\hat{\nabla}_{\Theta_t}\right\|_\infty \leq G_\infty$, the following inequality holds*

$$\sum_{t=1}^{T} \frac{\widehat{m}_{t,i}^2}{\sqrt{t\widehat{v}_{t,i}}} \leq \frac{2}{1 - \gamma} \frac{1}{\sqrt{1 - \beta_2}} \left\|\hat{\nabla}_{\Theta_{1:T,i}}\right\|_2.$$

The above lemma has been previously proven in Kingma & Ba (2014), and we omit its reproof here for brevity.

**Lemma A.4** *For a random matrix $\mathbf{X}$, the following inequality holds*

$$\mathbb{E}[\|\mathbf{X}\|_2] \leq \sqrt{\mathbb{E}[\|\mathbf{X}\|_2^2]}$$

*Proof.* According to the formula $\mathbb{E}[x^2] = \mathrm{Var}[x] + \mathbb{E}^2[x]$, we can derive:

$$\sqrt{\mathbb{E}[\|\mathbf{X}\|_2^2]} = \sqrt{\mathbb{E}^2[\|\mathbf{X}\|_2] + \mathrm{Var}[\|\mathbf{X}\|_2]}$$
$$\geq \sqrt{\mathbb{E}^2[\|\mathbf{X}\|_2]}$$
$$= \mathbb{E}[\|\mathbf{X}\|_2].$$

### A.1  Assumptions availability

**Bounded Parameters and Gradients.** It is reasonable to assume that parameters and gradients are bounded. This assumption is supported by Figure 2, which demonstrates the bounded nature of the gradients.

**Assumption on Bounded Gradients.** With bounded gradients, it follows that gradient variances and expectations are bounded (Assumption 4.1) and the gradient norms are also bounded (Assumption 4.4).

**Assumption on Bounded Parameters.** Given bounded parameters, the distance between parameters is naturally bounded (Assumption 4.2).

## A.2   THEOREM 4.3: CONVERGENCE OF SGD

*Proof.* The iteration form of SGD is

$$\boldsymbol{\Theta}_{t+1} \leftarrow \boldsymbol{\Theta}_t - \alpha_t \hat{\nabla}_{\boldsymbol{\Theta}_t}.$$

Subtract the scalar $\boldsymbol{\Theta}^*$ and square both sides of the above update, we have,

$$\|\boldsymbol{\Theta}_{t+1} - \boldsymbol{\Theta}^*\|^2 - \|\boldsymbol{\Theta}_t - \boldsymbol{\Theta}^*\|^2 = -2\alpha_t \hat{\nabla}_{\boldsymbol{\Theta}_t}(\boldsymbol{\Theta}_t - \boldsymbol{\Theta}^*) + \alpha_t^2 \hat{\nabla}_{\boldsymbol{\Theta}_t^2}.$$

Taking exception on both sides and use Assumption4.1, 4.2 and Lemma A.1, we have

$$\|\boldsymbol{\Theta}_{t+1} - \boldsymbol{\Theta}^*\|^2 - \|\boldsymbol{\Theta}_t - \boldsymbol{\Theta}^*\|^2 = -2\alpha_t \nabla_{\boldsymbol{\Theta}_t}(\boldsymbol{\Theta}_t - \boldsymbol{\Theta}^*) + \alpha_t^2 \mathbb{E}[\hat{\nabla}_{\boldsymbol{\Theta}_t^2}]$$

$$\leq -2\alpha_t \left[ \mathcal{L}_t(\boldsymbol{\Theta}_t) - \mathcal{L}_t(\boldsymbol{\Theta}^*) \right] + \alpha_t^2 \sum_{i=1}^{d} (\mathbb{E}[\hat{\nabla}_{\boldsymbol{\Theta}_{t,i}^2}])$$

$$\leq -2\alpha_t \left[ \mathcal{L}_t(\boldsymbol{\Theta}_t) - \mathcal{L}_t(\boldsymbol{\Theta}^*) \right] + \alpha_t^2 d(\sigma^2 + e^2).$$

Using $\alpha \geq \alpha_t$, we have

$$\|\boldsymbol{\Theta}_{t+1} - \boldsymbol{\Theta}^*\|^2 - \|\boldsymbol{\Theta}_t - \boldsymbol{\Theta}^*\|^2 \leq -2\alpha \left[ \mathcal{L}_t(\boldsymbol{\Theta}_t) - \mathcal{L}_t(\boldsymbol{\Theta}^*) \right] + \alpha^2 d(\sigma^2 + e^2)$$

Sum up for $t = 1, \ldots, T$,

$$\|\boldsymbol{\Theta}_{T+1} - \boldsymbol{\Theta}^*\|^2 - \|\boldsymbol{\Theta}_1 - \boldsymbol{\Theta}^*\|^2 \leq -2\alpha R^{SGD}(T) + \alpha^2 T d(\sigma^2 + e^2).$$

We can rearrange the above equation and $\|\boldsymbol{\Theta}_n - \boldsymbol{\Theta}_m\|_2 \leq D$,

$$R^{SGD}(T) \leq \frac{\|\boldsymbol{\Theta}_1 - \boldsymbol{\Theta}^*\|^2 - \|\boldsymbol{\Theta}_{T+1} - \boldsymbol{\Theta}^*\|^2}{2\alpha} + \frac{\alpha T d(\sigma^2 + e^2)}{2}$$

$$\leq \frac{D^2}{2\alpha} + \frac{\alpha T d(\sigma^2 + e^2)}{2}$$

## A.3   THEOREM 4.5: CONVERGENCE OF ADAM

*Proof.* The iteration of Adam is

$$\begin{cases} m_t = \beta_{1,t} \cdot m_{t-1} + (1 - \beta_{1,t}) \cdot \hat{\nabla}_{\boldsymbol{\Theta}_t}, \\ v_t = \beta_2 \cdot v_{t-1} + (1 - \beta_2) \cdot \left( \hat{\nabla}_{\boldsymbol{\Theta}_t} \right)^2, \\ \hat{m}_t = \frac{m_t}{1 - \beta_1^t}, \hat{v}_t = \frac{v_t}{1 - \beta_2^t} \\ \boldsymbol{\Theta}_{t+1} = \boldsymbol{\Theta}_t - \frac{\alpha}{\sqrt{\hat{v}} + \epsilon} \cdot \hat{m}_t. \end{cases}$$

Using Lemma A.1, we have,

$$\mathcal{L}_t(\boldsymbol{\Theta}_t) - \mathcal{L}_t(\boldsymbol{\Theta}^*) \leq \nabla_{\boldsymbol{\Theta}_t}^T (\theta_t - \theta^*) = \sum_{i=1}^{d} \nabla_{\boldsymbol{\Theta}_{t,i}} \left( \boldsymbol{\Theta}_{t,i} - \boldsymbol{\Theta}_{,i}^* \right).$$

From the above update rules presented, we have

$$\boldsymbol{\Theta}_{t+1} = \boldsymbol{\Theta}_t - \alpha_t \hat{m}_t / \sqrt{\hat{v}_t}$$

$$= \boldsymbol{\Theta}_t - \frac{\alpha_t}{1 - \beta_1^t} \left( \frac{\beta_{1,t}}{\sqrt{\hat{v}_t}} m_{t-1} + \frac{(1 - \beta_{1,t})}{\sqrt{\hat{v}_t}} \hat{\nabla}_{\boldsymbol{\Theta}_t} \right).$$

For the $i^{th}$ dimension of the parameter, we subtract the scalar $\boldsymbol{\Theta}_{,i}^*$ and square both sides of the above update rule, we have,

$$\left( \boldsymbol{\Theta}_{t+1,i} - \boldsymbol{\Theta}_{,i}^* \right)^2 = \left( \boldsymbol{\Theta}_{t,i} - \boldsymbol{\Theta}_{,i}^* \right)^2 - \frac{2\alpha_t}{1 - \beta_1^t} \left( \frac{\beta_{1,t}}{\sqrt{\hat{v}_{t,i}}} m_{t-1,i} + \frac{(1 - \beta_{1,t})}{\sqrt{\hat{v}_{t,i}}} \hat{\nabla}_{\boldsymbol{\Theta}_{t,i}} \right) \left( \boldsymbol{\Theta}_{t,i} - \boldsymbol{\Theta}_{,i}^* \right)$$

$$+ \alpha_t^2 \left( \frac{\hat{m}_{t,i}}{\sqrt{\hat{v}_{t,i}}} \right)^2.$$

We can rearrange the above equation and use Young's inequality, $ab \le a^2/2 + b^2/2$. Also, it can be shown that

$$\sqrt{\widehat{v}_{t,i}} = \sqrt{\sum_{j=1}^{t} (1 - \beta_2) \beta_2^{t-j} \hat{\nabla}_{\Theta_{j,i}^2} / \sqrt{1 - \beta_2^t}} \le \left\| \hat{\nabla}_{\Theta_{1:t,i}} \right\|_2, \tag{7}$$

and $\beta_{1,t} \le \beta_1$. Then

$$\hat{\nabla}_{\Theta_{t,i}} \left( \Theta_{t,i} - \Theta_{,i}^* \right) = \frac{(1 - \beta_1^t) \sqrt{\widehat{v}_{t,i}}}{2\alpha_t (1 - \beta_{1,t})} \left( \left( \Theta_{t,i} - \Theta_{,i}^* \right)^2 - \left( \Theta_{t+1,i} - \Theta_{,i}^* \right)^2 \right)$$

$$+ \frac{\beta_{1,t}}{(1 - \beta_{1,t})} \frac{\widehat{v}_{t-1,i}^{\frac{1}{4}}}{\sqrt{\alpha_{t-1}}} \left( \Theta_{,i}^* - \Theta_{t,i} \right) \sqrt{\alpha_{t-1}} \frac{m_{t-1,i}}{\widehat{v}_{t-1,i}^{\frac{1}{4}}}$$

$$+ \frac{\alpha_t (1 - \beta_1^t) \sqrt{\widehat{v}_{t,i}}}{2 (1 - \beta_{1,t})} \left( \frac{\widehat{m}_{t,i}}{\sqrt{\widehat{v}_{t,i}}} \right)^2$$

$$\le \frac{1}{2\alpha_t (1 - \beta_1)} \left( \left( \Theta_{t,i} - \Theta_{,i}^* \right)^2 - \left( \Theta_{t+1,i} - \Theta_{,i}^* \right)^2 \right) \sqrt{\widehat{v}_{t,i}}$$

$$+ \frac{\beta_{1,t}}{2\alpha_{t-1} (1 - \beta_{1,t})} \left( \Theta_{,i}^* - \Theta_{t,i} \right)^2 \sqrt{\widehat{v}_{t-1,i}}$$

$$+ \frac{\beta_1 \alpha_{t-1}}{2 (1 - \beta_1)} \frac{m_{t-1,i}^2}{\sqrt{\widehat{v}_{t-1,i}}} + \frac{\alpha_t}{2 (1 - \beta_1)} \frac{\widehat{m}_{t,i}^2}{\sqrt{\widehat{v}_{t,i}}}.$$

We apply Lemma A.3 to the above inequality and sum across all the dimensions for $i \in 1, \ldots, d$ and the iterations for $t \in 1, \ldots, T$:

$$\sum_{i=1}^{d} \sum_{t=1}^{T} \hat{\nabla}_{\Theta_{t,i}} \left( \Theta_{t,i} - \Theta_{,i}^* \right) \le \sum_{i=1}^{d} \frac{1}{2\alpha (1 - \beta_1)} \left( \Theta_{1,i} - \Theta_{,i}^* \right)^2 \sqrt{\widehat{v}_{1,i}}$$

$$+ \sum_{i=1}^{d} \sum_{t=2}^{T} \frac{1}{2 (1 - \beta_1)} \left( \Theta_{t,i} - \Theta_{,i}^* \right)^2 \left( \frac{\sqrt{\widehat{v}_{t,i}}}{\alpha_t} - \frac{\sqrt{\widehat{v}_{t-1,i}}}{\alpha_{t-1}} \right)$$

$$+ \frac{\alpha(1 + \beta_1) G_\infty}{(1 - \beta_1) \sqrt{1 - \beta_2}(1 - \gamma)^2} \sum_{i=1}^{d} \left\| \hat{\nabla}_{\Theta_{1:T,i}} \right\|_2$$

$$+ \sum_{i=1}^{d} \sum_{t=1}^{T} \frac{\beta_{1,t}}{2\alpha_t (1 - \beta_{1,t})} \left( \Theta_{,i}^* - \Theta_{t,i} \right)^2 \sqrt{\widehat{v}_{t,i}}$$

From the assumption, $\|\Theta_t - \Theta^*\|_2 \le D, \|\Theta_m - \Theta_n\|_\infty \le D_\infty$, we have

$$\sum_{i=1}^{d} \sum_{t=1}^{T} \hat{\nabla}_{\Theta_{t,i}} \left( \Theta_{t,i} - \Theta_{,i}^* \right) \le \frac{D^2}{2\alpha (1 - \beta_1)} \sum_{i=1}^{d} \sqrt{T \widehat{v}_{T,i}} + \frac{D_\infty^2}{2\alpha} \sum_{i=1}^{d} \sum_{t=1}^{T} \frac{\beta_{1,t}}{(1 - \beta_{1,t})} \sqrt{t \widehat{v}_{t,i}}$$

$$+ \frac{\alpha (1 + \beta_1) G_\infty}{(1 - \beta_1) \sqrt{1 - \beta_2}(1 - \gamma)^2} \cdot \sum_{i=1}^{d} \left\| \hat{\nabla}_{\Theta_{1:T,i}} \right\|_2$$

We apply Eq. 7 to the above inequality, we have

$$\sum_{i=1}^{d} \sum_{t=1}^{T} \hat{\nabla}_{\Theta_{t,i}} \left( \Theta_{t,i} - \Theta_{,i}^* \right) \le \frac{D^2 \sqrt{T}}{2\alpha (1 - \beta_1)} \sum_{i=1}^{d} \|\hat{\nabla}_{\Theta_{1:T,i}}\|_2 + \frac{D_\infty^2}{2\alpha} \sum_{i=1}^{d} \sum_{t=1}^{T} \frac{\beta_{1,t} \sqrt{t}}{(1 - \beta_{1,t})} \|\hat{\nabla}_{\Theta_{1:t,i}}\|_2$$

$$+ \frac{\alpha (1 + \beta_1) G_\infty}{(1 - \beta_1) \sqrt{1 - \beta_2}(1 - \gamma)^2} \sum_{i=1}^{d} \left\| \hat{\nabla}_{\Theta_{1:T,i}} \right\|_2$$

Take expectation on both sides of the above inequality and apply Lemma A.4, Assumption 4.1,

$$
\begin{aligned}
\sum_{i=1}^{d}\sum_{t=1}^{T}\nabla_{\boldsymbol{\Theta}_{t,i}}\left(\boldsymbol{\Theta}_{t,i}-\boldsymbol{\Theta}_{,i}^{*}\right) \leq & \frac{D^{2}\sqrt{T}}{2\alpha\left(1-\beta_{1}\right)}\sum_{i=1}^{d}\mathbb{E}[\|\hat{\nabla}_{\boldsymbol{\Theta}_{1:T,i}}\|_{2}] \\
& + \frac{\alpha\left(1+\beta_{1}\right)G_{\infty}}{\left(1-\beta_{1}\right)\sqrt{1-\beta_{2}}(1-\gamma)^{2}}\sum_{i=1}^{d}\mathbb{E}[|\hat{\nabla}_{\boldsymbol{\Theta}_{1:T,i}}\|_{2}] \\
& + \frac{D_{\infty}^{2}}{2\alpha}\sum_{i=1}^{d}\sum_{t=1}^{T}\frac{\beta_{1,t}\sqrt{t}}{(1-\beta_{1,t})}\mathbb{E}[\|\hat{\nabla}_{\boldsymbol{\Theta}_{1:t,i}}\|_{2}] \\
\leq & \frac{D^{2}\sqrt{T}}{2\alpha\left(1-\beta_{1}\right)}\sum_{i=1}^{d}\sqrt{\mathbb{E}[\|\hat{\nabla}_{\boldsymbol{\Theta}_{1:T,i}}\|_{2}^{2}]} \\
& + \frac{\alpha\left(1+\beta_{1}\right)G_{\infty}}{\left(1-\beta_{1}\right)\sqrt{1-\beta_{2}}(1-\gamma)^{2}}\sum_{i=1}^{d}\sqrt{\mathbb{E}[|\hat{\nabla}_{\boldsymbol{\Theta}_{1:T,i}}\|_{2}^{2}]} \\
& + \frac{D_{\infty}^{2}}{2\alpha}\sum_{i=1}^{d}\sum_{t=1}^{T}\frac{\beta_{1,t}\sqrt{t}}{(1-\beta_{1,t})}\sqrt{\mathbb{E}[\|\hat{\nabla}_{\boldsymbol{\Theta}_{1:t,i}}\|_{2}^{2}]} \\
\leq & \frac{D^{2}T}{2\alpha\left(1-\beta_{1}\right)}\sum_{i=1}^{d}\sqrt{\sigma^{2}+e^{2}} + \frac{\alpha\left(1+\beta_{1}\right)G_{\infty}\sqrt{T}}{\left(1-\beta_{1}\right)\sqrt{1-\beta_{2}}(1-\gamma)^{2}}\sum_{i=1}^{d}\sqrt{\sigma^{2}+e^{2}} \\
& + \frac{D_{\infty}^{2}}{2\alpha}\sum_{i=1}^{d}\sum_{t=1}^{T}\frac{\beta_{1,t}t}{(1-\beta_{1,t})}\sqrt{\sigma^{2}+e^{2}}.
\end{aligned}
$$

We can use arithmetic geometric series upper bound for the last term:

$$
\sum_{t=1}^{T}\frac{\beta_{1,t}}{(1-\beta_{1,t})}t \leq \sum_{t=1}^{T}\frac{1}{(1-\beta_{1})}\lambda^{t-1}t \leq \frac{1}{(1-\beta_{1})(1-\lambda)^{2}}
$$

Therefore, we have the following regret bound:

$$
\begin{aligned}
R(T) \leq & \sum_{i=1}^{d}\sum_{t=1}^{T}\nabla_{\boldsymbol{\Theta}_{t,i}}\left(\boldsymbol{\Theta}_{t,i}-\boldsymbol{\Theta}_{,i}^{*}\right) \\
\leq & \frac{((1-\lambda)^{2}D^{2}T+D_{\infty}^{2})d}{2\alpha\left(1-\beta_{1}\right)(1-\lambda)^{2}}\sqrt{\sigma^{2}+e^{2}} + \frac{\alpha\left(1+\beta_{1}\right)G_{\infty}\sqrt{T}d}{\left(1-\beta_{1}\right)\sqrt{1-\beta_{2}}(1-\gamma)^{2}}\sqrt{\sigma^{2}+e^{2}}
\end{aligned}
$$

# B  SUPPLEMENTARY PROOF OF THE THEORY

This section will demonstrate the convergence of Fully Quantized Training (FQT) in non-convex scenarios. The convergence of FQT can be expressed as:

$$
\mathbb{E}\left\|\nabla_{\boldsymbol{\Theta}}\right\|^{2}
$$

## B.1  ASSUMPTIONS

We assume:

**Assumption B.1** *The loss $\mathcal{L}(\boldsymbol{\Theta})$ is continuously differentiable and $\nabla_{\boldsymbol{\Theta}}$ is $\beta_{L}$-Lipschitz continuous.*

**Assumption B.2** *$\mathcal{L}(\boldsymbol{\Theta})$ is bounded below by $\mathcal{L}_{inf}$*

**Assumption B.3** *The variance of the gradient is bounded,i.e.,* $\mathrm{Var}\left[\hat{\nabla}_{\boldsymbol{\Theta}_{t,i}}\right] \leq \sigma^{2}$

## B.2 CONVERGENCE OF SGD

*Proof.* According to Assumption B.1, we have:

$$\left\| \nabla_{\boldsymbol{\Theta}_{t+1}} - \nabla_{\boldsymbol{\Theta}_t} \right\|_2 \leq \beta_L \left\| \boldsymbol{\Theta}_{t+1} - \boldsymbol{\Theta}_t \right\|_2$$

According to Bottou et al. (2018), we have

$$\mathcal{L}\left(\boldsymbol{\Theta}_{t+1}\right) - \mathcal{L}\left(\boldsymbol{\Theta}_t\right) \leq \nabla_{\boldsymbol{\Theta}_t}^\top \left(\boldsymbol{\Theta}_{t+1} - \boldsymbol{\Theta}_t\right) + \frac{1}{2}\beta_L \left\|\boldsymbol{\Theta}_{t+1} - \boldsymbol{\Theta}_t\right\|^2 \tag{8}$$

Plugging the SGD iteration, we have

$$\mathcal{L}\left(\boldsymbol{\Theta}_{t+1}\right) - \mathcal{L}\left(\boldsymbol{\Theta}_t\right) \leq -\alpha \nabla_{\boldsymbol{\Theta}_t}^\top \hat{\nabla}_{\boldsymbol{\Theta}_t} + \frac{1}{2}\alpha^2 \beta_L \left\|\hat{\nabla}_{\boldsymbol{\Theta}_t}\right\|^2$$

Taking expectations on both sides and applying AssumptionB.3,

$$\mathbb{E}\left[\mathcal{L}\left(\boldsymbol{\Theta}_{t+1}\right)\right] - \mathbb{E}\left[\mathcal{L}\left(\boldsymbol{\Theta}_t\right)\right] \leq -\alpha \left\|\nabla_{\boldsymbol{\Theta}_t}\right\|^2 + \frac{1}{2}\alpha^2 \beta_L \left(\mathrm{Var}\left[\hat{\nabla}_{\boldsymbol{\Theta}_t}\right] + \left\|\mathbb{E}\left[\hat{\nabla}_{\boldsymbol{\Theta}_t}\right]\right\|^2\right)$$

$$\leq -\alpha \left(1 - \frac{1}{2}\alpha\beta_L\right) \left\|\nabla_{\boldsymbol{\Theta}_t}\right\|^2 + \frac{1}{2}\alpha^2 \beta\sigma^2$$

$$\leq -\frac{1}{2}\alpha \left\|\nabla_{\boldsymbol{\Theta}_t}\right\|^2 + \frac{1}{2}\alpha^2 \beta_L \sigma^2$$

Summing the above equation up across iterations {1,...,T}, and utilize Assumption B.2, we have

$$L_{\mathrm{inf}} - \mathcal{L}\left(\boldsymbol{\Theta}_1\right) \leq \mathbb{E}\left[\mathcal{L}\left(\boldsymbol{\Theta}_{T+1}\right)\right] - \mathbb{E}\left[\mathcal{L}\left(\boldsymbol{\Theta}_1\right)\right] \leq -\frac{1}{2}\alpha \sum_{t=1}^{T} \mathbb{E}\left\|\nabla_{\boldsymbol{\Theta}_t}\right\|^2 + \frac{1}{2}T\alpha^2 \beta_L \sigma^2$$

Rearrange the terms, we have:

$$\mathbb{E}\left\|\nabla_{\boldsymbol{\Theta}_t}\right\|^2 \leq \frac{2\left(\mathcal{L}\left(\boldsymbol{\Theta}_1\right) - \mathcal{L}_{inf}\right)}{\alpha T} + \alpha\beta_L \sigma^2.$$

For $T \to \infty$, $\mathbb{E}\left\|\nabla_{\boldsymbol{\Theta}_t}\right\|^2 = O(\sigma^2)$.

## B.3 CONVERGENCE OF ADAM

*proof.* Substitute Adam's update rule into Eq. 8:

$$\mathcal{L}\left(\boldsymbol{\Theta}_{t+1}\right) - \mathcal{L}\left(\boldsymbol{\Theta}_t\right) \leq -\alpha \nabla_{\boldsymbol{\Theta}_t}^T \frac{\hat{m}_t}{\sqrt{\hat{v}_t} + \epsilon} + \frac{\beta_L}{2}\alpha^2 \left\|\frac{\hat{m}_t}{\sqrt{\hat{v}_t} + \epsilon}\right\|^2$$

Take the expectation of the first term, we have:

$$\mathbb{E}\left[-\alpha \nabla_{\boldsymbol{\Theta}_t}^T \frac{\hat{m}_t}{\sqrt{\hat{v}_t} + \epsilon}\right]$$

Since $\hat{m}_t$ is an unbiased estimate, we have

$$\mathbb{E}\left[-\alpha \nabla_{\boldsymbol{\Theta}_t}^T \frac{\hat{m}_t}{\sqrt{\hat{v}_t} + \epsilon}\right] = -\alpha \mathbb{E}\left[\frac{|\nabla_{\boldsymbol{\Theta}_t}|^2}{\sqrt{\hat{v}_t} + \epsilon}\right]$$

we have

$$\mathbb{E}\left[-\alpha \nabla_{\boldsymbol{\Theta}_t}^T \frac{\hat{m}_t}{\sqrt{\hat{v}_t} + \epsilon}\right] \leq -\alpha \mathbb{E}\left[\frac{|\nabla_{\boldsymbol{\Theta}_t}|^2}{\sqrt{\sigma^2 + \mathbb{E}[\nabla_{\boldsymbol{\Theta}_t}]^2}}\right].$$

Take the expectation of the second term, we have:

$$\mathbb{E}\left[\frac{\beta}{2}\alpha^2 \left|\frac{\hat{m}_t}{\sqrt{\hat{v}_t} + \epsilon}\right|^2\right] \leq \mathbb{E}\left[\frac{\beta_L}{2}\alpha^2 \frac{|\hat{m}_t|^2}{\sqrt{\hat{v}_t} + \epsilon}\right] = \mathbb{E}\left[\frac{\beta_L}{2}\alpha^2 \frac{|\nabla_{\boldsymbol{\Theta}_t}|^2}{\sqrt{\hat{v}_t} + \epsilon}\right]$$

Combining one and two, and substitute the definition of $\sqrt{\hat{v}_t}$ into:

$$\sqrt{\hat{v}_t} \leq \sqrt{\sigma^2 + \mathbb{E}\left[\nabla_{\boldsymbol{\Theta}_t}\right]^2}$$

we have:

$$\mathbb{E}[\mathcal{L}\left(\boldsymbol{\Theta}_{t+1}\right) - \mathcal{L}\left(\boldsymbol{\Theta}_t\right)] \leq -\alpha\left(1 - \frac{\beta_L}{2}\alpha\right)\mathbb{E}\left[\frac{|\nabla_{\boldsymbol{\Theta}_t}|^2}{\sqrt{\sigma^2 + \mathbb{E}[\nabla_{\boldsymbol{\Theta}_t}]^2}}\right]$$

Summing the above equation up across iterations {1,...,T}, and utilize Assumption B.2, we have

$$\mathcal{L}\left(\boldsymbol{\Theta}_1\right) - \mathcal{L}_{\inf} \geq \alpha\left(1 - \frac{\beta_L}{2}\alpha\right)\sum_{t=1}^{T}\mathbb{E}\left[\frac{|\nabla_{\boldsymbol{\Theta}_t}|^2}{\sqrt{\sigma^2 + \mathbb{E}[\nabla_{\boldsymbol{\Theta}_t}]^2}}\right]$$

Rearrange the terms, we have

$$\frac{1}{T}\sum_{t=1}^{T}\mathbb{E}\left[|\nabla_{\boldsymbol{\Theta}_t}|^2\right] \leq \frac{\mathcal{L}\left(\boldsymbol{\Theta}_1\right) - L_{\inf}}{\alpha T\left(1 - \frac{\beta_L}{2}\alpha\right)} \cdot \sqrt{\sigma^2 + \mathbb{E}[\nabla_{\boldsymbol{\Theta}_t}]^2}.$$

For $T \to \infty$, $\mathbb{E}\left\|\nabla_{\boldsymbol{\Theta}_t}\right\|^2 = O(\sigma)$.

## B.4 CONVERGENCE OF SGD-M

The iteration of SGD-M is

$$v_{t+1} = \beta v_t + (1 - \beta)\hat{\nabla}_{\boldsymbol{\Theta}_t}$$
$$\boldsymbol{\Theta}_{t+1} = \boldsymbol{\Theta}_t - \alpha v_{t+1}$$

*proof.* Substitute SGD-M's update rule into Eq. 8:

$$\mathcal{L}\left(\boldsymbol{\Theta}_{t+1}\right) - \mathcal{L}\left(\boldsymbol{\Theta}_t\right) \leq -\alpha\nabla_{\boldsymbol{\Theta}_t}^T v_{t+1} + \frac{\beta_L}{2}\alpha^2\left\|v_{t+1}\right\|^2$$

Take the expectation of the first term, we have:

$$\mathbb{E}\left[-\alpha\nabla_{\boldsymbol{\Theta}_t}^T v_{t+1}\right] = -\alpha\mathbb{E}\left[|\nabla_{\boldsymbol{\Theta}_t}|^2\right]$$

Take the expectation of the second term, we have

$$\frac{\beta_L}{2}\alpha^2\mathbb{E}[|v_{t+1}|^2] \leq \frac{\beta_L}{2}\alpha^2\left(\frac{\mathbb{E}[|\nabla_{\boldsymbol{\Theta}_t}|^2]}{1 - \beta^2} + \frac{\sigma^2}{1 - \beta^2}\right).$$

Combining one and two, we have

$$\mathcal{L}\left(\boldsymbol{\Theta}_{t+1}\right) - \mathcal{L}\left(\boldsymbol{\Theta}_t\right) \leq -\alpha(1 - \frac{\beta_L\alpha}{2(1 - \beta^2)})\mathbb{E}[|\nabla_{\boldsymbol{\Theta}_t}|^2] + \frac{\beta_L\alpha^2\sigma^2}{2(1 - \beta^2)}$$

Summing the above equation up across iterations {1,...,T}, and utilize Assumption B.2, we have

$$\mathcal{L}\left(\boldsymbol{\Theta}_1\right) - \mathcal{L}_{\inf} \geq \sum_{t=1}^{T}[\alpha(1 - \frac{\beta_L\alpha}{2(1 - \beta^2)})\mathbb{E}[|\nabla_{\boldsymbol{\Theta}_t}|^2]] - T\frac{\beta_L\alpha^2\sigma^2}{2(1 - \beta^2)}$$

Rearrange the terms, we have

$$\frac{1}{T}\sum_{t=1}^{T}\mathbb{E}\left[|\nabla_{\boldsymbol{\Theta}_t}|^2\right] \leq \frac{\mathcal{L}\left(\boldsymbol{\Theta}_1\right) - L_{\inf}}{T\alpha(1 - \frac{\beta_L\alpha}{2(1-\beta^2)})} + \frac{\beta_L\alpha\sigma^2}{2(1 - \beta^2)(1 - \frac{\beta_L\alpha}{2(1-\beta^2)})}.$$

For $T \to \infty$, $\mathbb{E}\left\|\nabla_{\boldsymbol{\Theta}_t}\right\|^2 = O(\sigma^2)$.

## C  VARIANCE OF SPECIFIC QUANTIZERS

**Proposition C.1** *(Variance of stochastic rounding) For any number $X \in \mathbb{R}$, $\mathrm{Var}[\mathrm{SR}(X)] \leq \frac{1}{4}$.*

*Proof.* For any real number $X$, let $p := X - \lfloor X \rfloor \in [0, 1)$, then

$$\mathrm{Var}[\mathrm{SR}(X)] = \mathbb{E}[\mathrm{SR}(X) - X]^2 = p(\lceil X \rceil - X)^2 + (1 - p)(\lfloor X \rfloor - X)^2$$

$$= p(1 - p)^2 + p^2(1 - p) = p(1 - p)(1 - p + p) = p(1 - p) \leq \frac{1}{4}.$$

### C.1  PER-SAMPLE QUANTIZER

Given an activation gradient $\hat{\nabla}_{\mathbf{H}^{(l)}}$, its per-sample quantization is:

$$Q_g(\hat{\nabla}_{\mathbf{H}_{i,j}^{(l)}}) = \mathrm{SR}(B(\hat{\nabla}_{\mathbf{H}_{i,j}^{(l)}} - Z_i)/R_i)R_i/B + Z_i,$$

where apply different ranges $R_i$ and zero points $Z_i$ for each sample of the gradient. When $\mathbf{S} = \mathrm{diag}\left\{\frac{R_1}{B}, \ldots, \frac{R_N}{B}\right\}$, we have

$$\mathrm{Var}\left[Q_g\left(\hat{\nabla}_{\mathbf{H}^{(l)}}\right)\right] = \mathrm{Var}\left[\mathbf{S}\,\mathrm{SR}\left((\mathbf{S}^{-1}\left(\hat{\nabla}_{\mathbf{H}^{(l)}} - \mathbf{1Z}\right))/ + \mathbf{1Z}\right]$$

$$\leq \sum_{i=1}^{N}\sum_{j=1}^{D^{(l)}} \mathrm{Var}[\frac{R_i}{B}\,\mathrm{SR}(\frac{B}{R_i}(\hat{\nabla}_{\mathbf{H}_{i,j}^{(l)}} - Z_i)) + Z_i)]$$

$$= \sum_{i=1}^{N}\sum_{j=1}^{D^{(l)}} \frac{R_i^2}{B^2}\,\mathrm{Var}[\mathrm{SR}(\frac{B}{R_i}(\hat{\nabla}_{\mathbf{H}_{i,j}^{(l)}} - Z_i))]$$

$$\leq \frac{D^{(l)}}{4B^2}\sum_{i=1}^{N} R_i^2.$$

### C.2  PER-SAMPLE QUANTIZER WITH AGP

Place the groups with the largest range in the first $N/b$ rows, and let the range of these groups be denoted by $R_1, \ldots, R_{N/b}$, groups in the remaining rows are denoted by $r_{N/b+1}, \ldots, r_N$. We assume that $r/R \approx 0$.

$$Q_g(\hat{\nabla}_{\mathbf{H}^{(l)}}) = (\mathbf{MS})\,\mathrm{SR}\left((\mathbf{MS})^{-1}\left(\mathbf{M}\hat{\nabla}_{\mathbf{H}^{(l)}} - \mathbf{MZ}\right)\right) + \mathbf{MZ},$$

where $\mathbf{M} = \mathrm{diag}\left(\frac{m_1}{p_1}, \ldots, \frac{m_N}{p_N}\right)$, $p_i = \frac{NR_i}{bR_{total}}$, $R_{total} = \sum_{i=1}^{N} R_i$ and $m_i \sim \mathrm{Bern}(p_i)$. To simplify the problem, we assume that $R_1 \approx R_2 \cdots \approx R_{N/b}$. And we use $r/R \approx 0$, then $p \approx \{1, \ldots, 0\}$. In other words, for the first $\frac{N}{b}$ rows, $m = 1$, and 0 otherwise. We substitute it into the above equation and prune the groups with smaller ranges,

$$Q_g(\hat{\nabla}_{\mathbf{H}^{(l)}}) = \mathbf{S}_{1:\frac{N}{b},1:\frac{N}{b}}\,\mathrm{SR}\left((\mathbf{S}_{1:\frac{N}{b},1:\frac{N}{b}})^{-1}\left(\hat{\nabla}_{\mathbf{H}_{1:\frac{N}{b}}^{(l)}} - \mathbf{1Z}_{1:\frac{N}{b}}\right)\right) + \mathbf{1Z}_{1:\frac{N}{b}}.$$

Then we have,

$$\mathrm{Var}\left[Q_g\left(\hat{\nabla}_{\mathbf{H}^{(l)}}\right)\right] \leq \sum_{i=1}^{N/b}\sum_{j=1}^{D^{(l)}} \mathrm{Var}[\frac{R_i}{B}\,\mathrm{SR}(\frac{B}{R_i}(\hat{\nabla}_{\mathbf{H}_{i,j}^{(l)}} - Z_i)) + Z_i)]$$

$$= \sum_{i=1}^{N/b}\sum_{j=1}^{D^{(l)}} \frac{R_i^2}{B^2}\,\mathrm{Var}[\mathrm{SR}(\frac{B}{R_i}(\hat{\nabla}_{\mathbf{H}_{i,j}^{(l)}} - Z_i))]$$

$$\leq \frac{D^{(l)}}{4B^2}\sum_{i=1}^{N/b} R_i^2.$$

For 1-bit quantizers, the variance of PSQ is $\frac{D^{(l)}}{4}(\sum_{i=1}^{N/b} R_i^2 + \sum_{i=N/b+1}^{N} r_i^2)$. It is clear that

$$\frac{D^{(l)}}{4B^2}\sum_{i=1}^{N/b} R_i^2 \leq \frac{D^{(l)}}{4}(\sum_{i=1}^{N/b} R_i^2 + \sum_{i=N/b+1}^{N} r_i^2).$$

## D  IMPLEMENTATION DETAILS

We implemented our method as a lightweight library in PyTorch. For binary matrix multiplication and some auxiliary operations, we implemented them using C++. In Alg. 1, we illustrate the process of forward and backward propagation for quantized fully connected layers. For simplicity, certain details, such as bias terms, quantization zero points, and the splitting operations on gradient tensors, are omitted here. The entire process primarily consists of five components: quantization (9), encoding(4-5, 10), low-bit multiplication (6, 11), pruning (8), and dequantization (12).

---

**Algorithm 1** Linear Layer Forward and Backward Propagation

---

1: **Input:** Input $\mathbf{H}^{(l-1)}$, Weight $\mathbf{\Theta}^{(l)}$, Gradient of Loss $\nabla_{\mathbf{H}^{(l)}}$
2: **Output:** Output $\mathbf{H}^{(l)}$, Gradient of Weight $\nabla_{\mathbf{\Theta}^{(l)}}$, Gradient of Input $\nabla_{\mathbf{H}^{(l-1)}}$
3: // Forward Propagation
4: Encode Weight: $\overline{\overline{\mathbf{H}}}^{(l-1)} = \text{row\_encoder}(\mathbf{H}^{(l-1)})$
5: Encode Input: $\overline{\mathbf{\Theta}}^{(l)} = \text{column\_encoder}(\mathbf{\Theta}^{(l)})$
6: Compute Output: $\mathbf{H}^{(l)} = \overline{\overline{\mathbf{H}}}^{(l-1)}\overline{\mathbf{\Theta}}^{(l)}$
7: // Backward Propagation
8: Pruning: $\nabla_{\mathbf{H}_{PSQ}^{(l)}}, \nabla_{\mathbf{H}_{PCQ}^{(l)}} = \text{pruner}(\nabla_{\mathbf{H}^{(l)}})$
9: Quantization: $\overline{\nabla}_{\mathbf{H}_{PSQ}^{(l)}}, \mathbf{S}_{PSQ}^{(l)} = \text{PSQ}(\nabla_{\mathbf{H}_{PSQ}^{(l-1)}})$,

 $\overline{\nabla}_{\mathbf{H}_{PCQ}^{(l)}}, \mathbf{S}_{PCQ}^{(l)} = \text{PCQ}(\nabla_{\mathbf{H}_{PCQ}^{(l-1)}})$
10: Encode Gradient: $\overline{\overline{\nabla}}_{\mathbf{H}_{PSQ}^{(l)}} = \text{row\_encoder}(\overline{\nabla}_{\mathbf{H}_{PSQ}^{(l)}})$,

 $\overline{\overline{\nabla}}_{\mathbf{H}_{PCQ}^{(l)}} = \text{column\_encoder}(\overline{\nabla}_{\mathbf{H}_{PCQ}^{(l)}})$
11: Compute Gradient: $\overline{\nabla}_{\mathbf{\Theta}^{(l)}} = \overline{\overline{\mathbf{H}}}^{(l-1)^{\top}}\overline{\nabla}_{\mathbf{H}_{PCQ}^{(l)}}$,

 $\overline{\nabla}_{\mathbf{H}^{(l-1)}} = \overline{\nabla}_{\mathbf{H}_{PSQ}^{(l)}}\overline{\mathbf{\Theta}}^{(l)^{\top}}$
12: Dequantization: $\hat{\nabla}_{\mathbf{\Theta}^{(l)}} = \overline{\nabla}_{\mathbf{\Theta}^{(l)}}\mathbf{S}_{PCQ}^{(l)}$, $\hat{\nabla}_{\mathbf{H}^{(l-1)}} = \mathbf{S}_{PSQ}^{(l)}\overline{\nabla}_{\mathbf{H}^{(l-1)}}$
13: // Update Parameters
14: Update Weight: $\mathbf{W} \leftarrow \mathbf{W} - \alpha\hat{\nabla}_{\mathbf{\Theta}^{(l)}}$

---

**Encoder** is a functional component that encodes multiple integers with values of 1 or -1 into a smaller set of elements, facilitating subsequent XNOR operations. Taking Row_Encoder as an example, its primary form is illustrated in Alg. 2, the case where the number of columns is not divisible by $b$ has been overlooked.

**Binary multiplication** is the crucial operation. In our approach, both forward and backward propagation are implemented through binary multiplication. For example, For two vectors, $\mathbf{X}_1$ and $\mathbf{X}_2$, each of length 32, encode them into two unsigned 32-bit integers, $x_1$ and $x_2$. The multiplication of the two is implemented as follows:

$$\text{SUM}(\mathbf{X}_1 \odot \mathbf{X}_2) = \text{bitcount}(\text{XNOR}(x1, x2)) << 1 - 32$$

where the dot product of two vectors, each of length 32, is efficiently replaced by a single bitcount and XNOR operation, effectively reducing energy consumption and time overhead. However, it is worth noting that an unbiased quantizer maps data to 0 or 1, rather than -1 or 1. Therefore, some conversion is required. For $\mathbf{X}_1 \in \{1, -1\}^n$, $\mathbf{X}_2 = \text{ReLU}(\mathbf{X}_1)$, it is clear that $(S/2)\mathbf{X}_1 + Z + (S/2) = S\mathbf{X}_2 + Z$. Therefore, some adjustments are needed: a straightforward modification of the scaling factor $S$ and zero point $Z$ is sufficient to achieve the transformation. This transformation requires only one multiplication and one addition for the scaling factor and zero point, thus incurring minimal overhead.

---

**Algorithm 2** Row_Encoder

---

1: **Input:** Input $\mathbf{H} \in \mathbb{R}^{N \times D}$, Bits $b$
2: **Output:** Output $\mathbf{H}_e \in \mathbb{R}^{N \times \lfloor (1+(D-1)/b) \rfloor}$
3: **for** $i \leftarrow 1$ to $N$ **do**
4:     **for** $j \leftarrow 1$ to $\lfloor (1+(D-1)/b) \rfloor$ **do**
5:         $\mathbf{H}_{i,j} = 0$
6:         **for** $k \leftarrow 1$ to $b$ **do**
7:             $s = (\mathbf{H} > 0)$
8:             $\mathbf{H}_{i,j} = (\mathbf{H}_{i,j} << 1) \| s$
9:         **end for**
10:     **end for**
11: **end for**

---

Table 7: Comparison of computational operations for multiplying matrices of size $N \times D$ and $D \times D^{(l-1)}$ in 1-bit matrix multiplication (MM).

| Setting | XNOR | BitCount | Shift | FP Addition | INT Addition | FP Multiplication | AND |
|---|---|---|---|---|---|---|---|
| 1-bit MM | $ND^{(l-1)}D$ | $(N+1)D^{(l-1)}D$ | - | $ND^{(l-1)}$ | - | $2ND^{(l-1)}$ | - |
| Average 1-bit MM | $ND^{(l-1)}D$ | $(N+1)D^{(l-1)}D$ | $N(\frac{3}{4}D + \frac{3}{2}D^{(l-1)})$ | $\frac{N}{4}D^{(l-1)}$ | $\frac{3}{4}ND^{(l-1)}$ | $\frac{1}{2}ND^{(l-1)}$ | $ND$ |

For convolutional layers, direct matrix multiplication is not feasible. To facilitate subsequent operations, an unfolder is performed on the convolutional layer before matrix multiplication. After computation, the standard form is restored through folder operations. For example, We perform a convolution operation between the input $\mathbf{X} \in \mathbb{R}^{N \times C \times H \times W}$ and parameters $\boldsymbol{\Theta} \in \mathbb{R}^{D \times C \times K \times K}$ to obtain the output $\mathbf{Y} \in \mathbb{R}^{N \times D \times H \times W}$.

**1-bit Matrix Multiplication vs. Average 1-bit Matrix Multiplication.** The main difference between standard 1-bit MM and average 1-bit MM is that the latter introduces Shift, INT Addition, and AND operations due to matrix splitting and summing four submatrices. Specifically, average 1-bit MM incurs $N(\frac{3}{4}D + \frac{3}{2}D^{(l-1)})$ Shift operations, $\frac{3}{4}ND^{(l-1)}$ INT Addition operations, and $ND$ AND operations. However, these operations are relatively few and lightweight, so they do not significantly increase the time cost.

**Unfolder** treats each element involved in element-wise multiplication within the kernel as a row, and the number of times the window slides as columns, the unfolded input and parameters transform into $\mathbf{X}_u \in \mathbb{R}^{NHW \times CK^2}$, $\boldsymbol{\Theta}_u \in \mathbb{R}^{D \times CK^2}$. Finally, we need to restore the output $\mathbf{Y}_u \in \mathbb{R}^{NHW \times D}$ to its standard state.

**Folder** is the inverse operation of Unfolder, designed to restore the gradients of both the input and parameters $\nabla_{\mathbf{X}_u} \in \mathbb{R}^{NHW \times CK^2}$, $\nabla_{\boldsymbol{\Theta}_u} \in \mathbb{R}^{D \times CK^2}$ to their standard states.

# E EXPERIMENTAL DETAILS

## E.1 GRADIENT DISTRIBUTION

From Fig. 6, it can be observed that the gradient of the activation exhibits a pattern across different epochs: the ranges of groups (both samples and dimensions) are highly uneven. Some groups have large ranges, while others have small ranges. Although we have presented results for a single batch, the same pattern persists across the remaining batches.

## E.2 EXPERIMENTAL SETTINGS

**Classification task:** The training process is divided into two stages: initially undergoing quantization-aware training on ImageNet and subsequently undergoing FQT on various downstream datasets. The first stage: the initial learning rate was set to $10^{-3}$ and the weight decay to $10^{-5}$, following Bulat & Tzimiropoulos (2019), the optimizer is Adam and use a consine learning rate schedule. We train for 90 epochs. The second stage: for all datasets, the initial learning rate for fully connected layers is set to $10^{-3}$. For portions of the network that have been previously trained, the learning rate is set to $10^{-5}$,

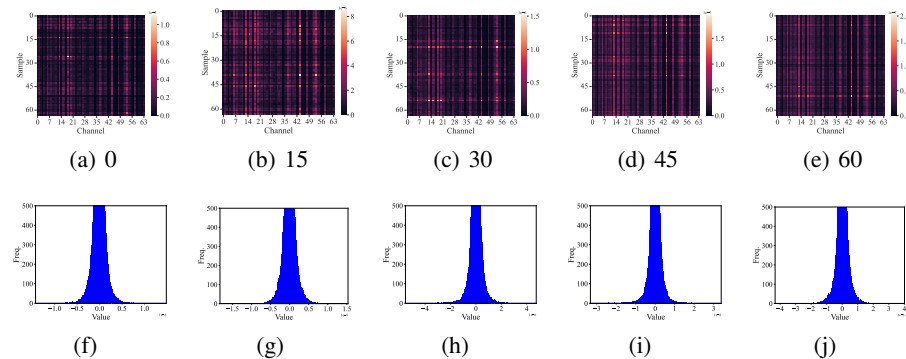

Figure 6: Heterogeneity in a ResNet18's gradients. (a-e) Heatmap of the per-group range at the conv2.1.2 layer under different epochs; (f-j) Histogram of the gradient groups (samples) at the same layer.

Table 8: Validation loss of fine-tuning GPT2-xl on the Shakespeare dataset.

| Method | iter0 | iter5 | iter10 | iter15 | iter20 |
|---|---|---|---|---|---|
| Full Precision Training | 3.76 | 2.82 | 2.81 | 2.88 | 2.85 |
| Ours | 3.76 | 3.37 | 3.32 | 3.38 | 3.24 |
| PSQ | 3.76 | nan | nan | nan | nan |

except for car dataset (Krause et al., 2013) where it is set to $10^{-4}$. The optimizer settings are the same as the first stage. We train for 60 epochs. The batch size was assigned to be 128. We measured training latency on CPUs, but to expedite the acquisition of accuracy statistics, we simulated the training results on 4 NVIDIA RTX A4000 GPUs. Due to limited resources on terminal devices, we utilized a smaller batch size of 64. We followed the configuration of Bulat & Tzimiropoulos (2019) by excluding quantization for sensitive layers, such as the first and last layers, as well as skip connections in residual networks, in addition to batch normalization (BN) and ReLU layers.

**Detection task:** We evaluate our method on a simple transfer learning task to assess its effectiveness on object detection models, specifically transferring from high-resolution object detection to low-resolution object detection. The training process is divided into two stages: initially undergoing quantization-aware training on the PASCAL VOC 2007 and VOC 2012 trainval sets with a resolution of (600*600) pixels, followed by FQT training on the same dataset with a resolution of (300*300) pixels. The first stage: We followed all the settings of BiDet (Wang et al., 2020), including the quantization methods for both weights and activation values and training configurations. The batch size was assigned to 32, and the Adam optimizer was applied. The learning rate started from $10^{-3}$ and dropped during training every 6 epochs by a factor of 10. We train for 20 epochs. The second stage: the initial learning rate is $10^{-5}$, the training epoch is 5 and the others are the same.

**NLP tasks:** We conduct experiments to validate the effectiveness of our proposed 1-bit FQT on BERT$_{\text{BASE}}$(12 hidden layers) and the GLUE benchmark (Wang et al., 2018a) which consists of nine basic language tasks. We use the standard metrics for each GLUE task to evaluate our method. We use Spearman Correlation for STS-B, Mathews Correlation Coefficient for CoLA, and classification accuracy for the rest tasks. As for the MNLI task, we report the accuracy on both in-domain evaluation MNLI-match (MNLI-m) and cross-domain evaluation MNLI-mismatch (MNLI-mm). We exclude the WNLI task as Qin et al. (2022). We utilized BiBERT(Qin et al., 2022) as our binarized model, which is derived by directly binarizing a full-precision one. Subsequently, we fine-tune this binarized model using both full-precision gradients (QAT) and 1-bit gradients (Ours). We follow Qin et al. (2022) by excluding quantization for the classifier, position embedding layer, and token type embedding layer. We use Adam as our optimizer. The training settings are also the same as Qin et al. (2022).

Table 9: Experimental results of training from scratch.

| Method | Precision (W, A, G) | XNOR-Net++ | | | Adabin | | |
|--------|---------------------|-----------|-----------|----------|----------|-----------|----------|
| | | CIFAR-10 | CIFAR-100 | ImageNet | CIFAR-10 | CIFAR-100 | ImageNet |
| QAT | 1, 1, 32 | 84.31 | 62.14 | 57.10 | 91.91 | 66.68 | 62.50 |
| PSQ | 1, 1, 1 | 24.27 | 6.14 | 0.10 | 41.59 | 16.09 | 0.10 |
| Ours | 1, 1, 1 | 42.57 | 26.96 | 21.63 | 78.80 | 58.14 | 39.12 |

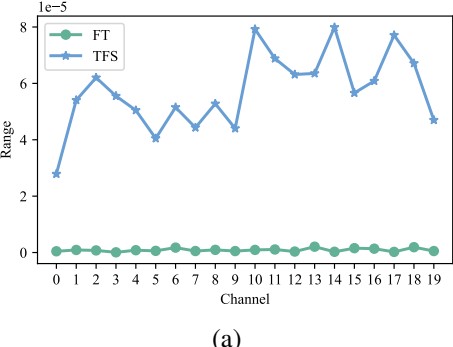 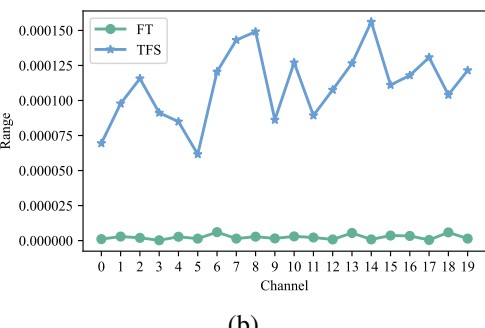

(a) (b)

Figure 7: Gradient range analysis in ResNet18's conv2.1.2 layer under fine-tuning (FT) and training from scratch (TFS). (a) The result from CIFAR-10. (b) The result from CIFAR-100.

### E.3 FQT from scratch

We compared the performance of our method in two scenarios: fine-tuning and training from scratch. We presented the classification results under the optimal configuration (b=4) in Table 9. From the table, it is evident that when training from scratch, the model exhibits very low classification accuracy across all datasets, and in certain datasets, it even lacks classification capability entirely. We attempted to analyze the differences between the two scenarios based on the distinct gradient distributions. From Fig. 7, we observe that the gradient range is larger in training from scratch, leading to increased gradient variance (Eq. 5) and reduced model convergence. Therefore, 1-bit FQT from scratch remains an open problem. Additionally, we compared our method with PSQ in the training scenario from scratch, and the results indicate that our approach still significantly outperforms PSQ in accuracy.

### E.4 Time expenditure structure

We present the speedup across layers of VGGNet16 and the time consumption for each operation in Fig. 10, providing guidance for future optimization directions. It is important to note that the first and last layers were not quantized and, therefore, were not included in the analysis. From the figure, it is evident that matrix multiplication constitutes the majority of the training time, while the time overhead of other operations such as gradient pruning and quantization can be considered negligible. Therefore, the focus of future optimization efforts will remain on matrix multiplication. Furthermore, it can be observed that our implemented method is particularly friendly for layers with a large number of convolutional kernels and smaller input resolution.

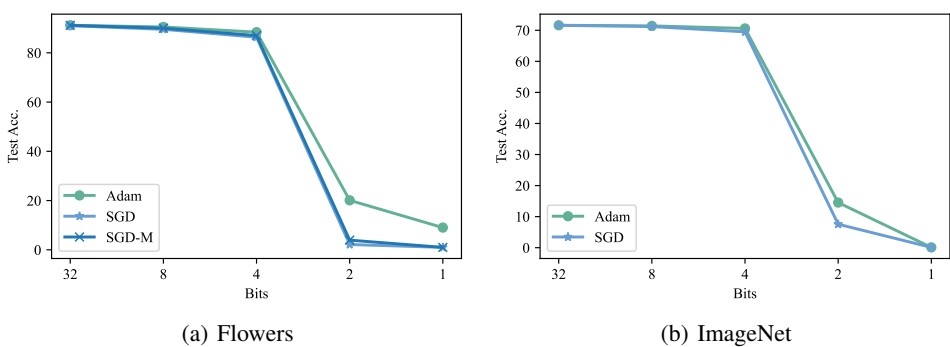

(a) Flowers       (b) ImageNet

Figure 8: Gradient numerical precision("bits") vs. test accuracy of VGGNet16 on (a) Flowers and (b) ImageNet.

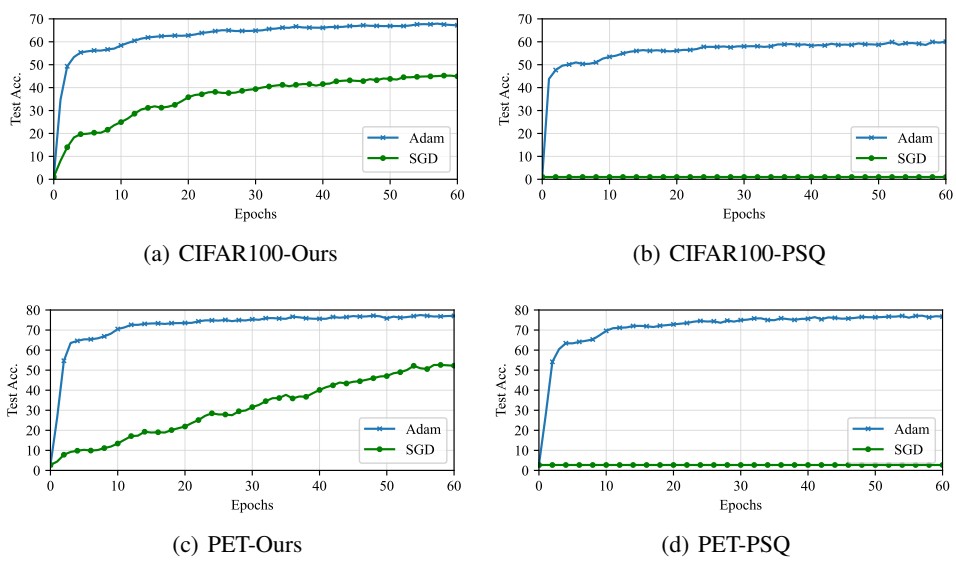

(a) CIFAR100-Ours      (b) CIFAR100-PSQ

(c) PET-Ours       (d) PET-PSQ

Figure 9: Testing accuracy comparison on VGGNet16.

Table 10: Experimental results on multiple downstream datasets. "(W, A, G)" denote the bitwidth of weight, activations, and gradients, respectively. $b$ represents the bitwidth of the remaining groups.

| Method | Precision (W, A, G) | Accuracy(%) CIFAR-10 | CIFAR-100 | Flowers | Cars | Pets | CUB | Average |
|---|---|---|---|---|---|---|---|---|
| | | ResNet-50 | | | | | | |
| QAT | 1, 1, 32 | 90.12 | 70.3 | 85.69 | 58.62 | 78.30 | 48.03 | 71.84 |
| PSQ | 1, 1, 1 | 77.85 | 54.39 | 84.61 | 34.52 | 76.75 | 42.34 | 61.74 |
| Ours ($b = 4$) | 1, 1, 1 | **82.84** | **60.19** | **85.49** | **47.08** | **77.86** | **45.20** | **66.44** |

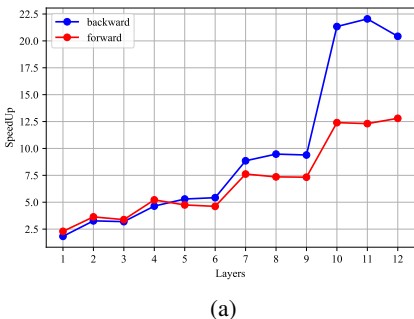
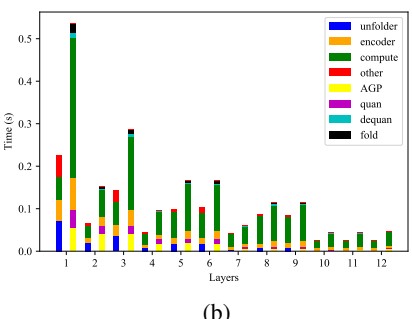

(a)                              (b)

Figure 10: (a) The speedup of ours compared with FP32 PyTorch. (b) The compositional structure of time consumption.

