# OpenReview forum: "1-Bit FQT: Pushing the Limit of Fully Quantized Training to 1-bit"
_ICLR.cc/2025/Conference — Submitted to ICLR 2025_

### Official Review · Reviewer_cWvz · 2024-10-20

**Soundness:** 2
**Presentation:** 2
**Contribution:** 1
**Rating:** 5
**Confidence:** 4

**Summary:**

The paper aims to accelerate training by applying binary quantization to the weights, activations, and gradients, making all three matrix multiplications binary and thus accelerating the process. This approach differs from standard Quantization-Aware Training (QAT), where typically only the weights and activations are quantized.

The paper also analyzes the variance introduced by gradient quantization when using both Adam and SGD optimizers. Based on this analysis, it proposes two methods for binary quantization:
1. Activation gradient pruning, which removes less informative gradient groups to improve numerical precision in the remaining groups.
2. Sample-channel joint quantization, which applies different quantization strategies to the gradients of weights and activations.

**Strengths:**

The paper introduce an interesting analysis of the difference between SGD and Adam optimizer under binary quantization.

**Weaknesses:**

I see several weaknesses in the paper, with the primary one being the experiments section, which I find very underdeveloped. Presenting results solely on small datasets like CIFAR, especially in the current era of LLMs, is not sufficient for a conference like ICLR. Additionally, the networks used, such as VGG and ResNet18, are quite outdated.

Even with these small datasets and older networks, the paper shows a significant performance gap compared to the baseline models, raising doubts about the approach's viability in real-world applications. The authors need to put more effort into this section to make the work more convincing.

Furthermore, the practical acceleration claims are also problematic. The approach requires non-trivial preprocessing of data for every General Matrix Multiply (GEMM) operation, making it challenging to support natively in hardware

**Questions:**

1) Do you try to run the proposed method on modern LLMs - opensource githubs like nanoGPT is a good baseline.
2) Is the speedup presented in Fig8 and table include all the preprocessing require in AQ when b > 1  - or it include only the matrix multiplication (which is unfair comparison)?   Can you also add comparison with BF16 which is usually ~2X and full time to train comparison.


======================================================

Dear Authors,

Thank you for your efforts in addressing the questions and concerns raised.
I decided to raise my score to 5 - still under the acceptance threshold.
I feel that the experiments still require more effort- mainly in larger dataset and more modern models.

---

### Official Review · Reviewer_M3f1 · 2024-10-28

**Soundness:** 3
**Presentation:** 3
**Contribution:** 3
**Rating:** 6
**Confidence:** 4

**Summary:**

First method to propose FQT where weight, activation and gradients are computed using 1 bit. The models have (average) 5% lower accuracy on classification datasets compared to 1 bit FQT using full precision gradients and less than 1% in simple datasets like Flowers and Pets dataset with an average speedup of 5.13x (training speedup) compared to FP32. The approach is explained with solid mathematical foundation and results are shown for both classification as detection models (thought only for fine-tuning in some cases). The 1 bit FTQ has a huge potential use case in memory restricted devices as the memory saving can be 32x (theoretically).

**Strengths:**

- One of the first work to show end to end training using 1 bit which includes gradients and showed results on models like ResNet18 and VGG-16
- Mathematical foundation of the approach is solid and reasonable (with few assumptions which may not hold true always)
- Finetuning using 1 bit FQT shows promising results compared to similar QAT methods
- 1 bit FTQ practical benefits is shows by using X-NOR 1 bit operations to save memory as well as computation.

**Weaknesses:**

- Memory saving analysis using 1 bit FTQ would add a lot of value to this method. Practically this method can be used to fine tune models running to tiny devices with very small memory and latency is not a prime concern for those devices.
- The technique should be tested on deeper model to see the impact of vanishing gradient problem. This issue may be more prominent when gradient is calculated in lower precision.
- Although L083 mentions that primary aim of this paper is to explore the ultimate limit of quantization instead of focussing on practical application performance, the usefulness of this method for practical use case can strengthen the paper.
- The training speedup of 5.13x is compared with FP32 training, the Author(s) are advised to also provide speedup as well as accuracy numbers compared to full INT8 to see the extent of benefits of 1 bit FQT for real use cases.
- The approach is tested on very old models like ResNet18 and VGG16, more results on other challenging but faster models will be interesting to see (eg. Mobilenet)
- 1 bit FQT is only applicable to transfer learning and puts a big question on applicability of this approach.

**Questions:**

- One of the initial assumption that Adam is better in low precision regime is based on the sensitivity analysis in Figure 1. However, since it is the foundational assumption of the paper (L235, L221) and I would recommend to add results for more complex datasets as CiFAR10 is too simple and other datasets may change this assumption (mid size like Flowers/Pascal VOC and large size like Image-net/coco). Although this is also proved mathematically (L221) under certain assumptions (4.1 to 4.4), it would be good to verify these on other datasets
- L270: Average 1-bit Quantization (AQ) - Is it similar to mixed precision quantization where some of the groups are quantized with higher precision (2, 4 bits etc)? If yes, what is there any information of percentage of 1-bit, other bits and %age of pruned channels for any dataset/model available ?
- Table 1, L385, L386 : what is the percentage of “remaining groups” using b bandwidth?
- L385, L386 : intuitively the accuracy of models where “remaining groups” use 8 bit should have more accuracy but Table 1 suggests otherwise. This is mentioned in L409 but adding intuition behind this would be helpful (confirmed in L426)
- Effect of optimizer (L462) - The experiment may need to be expanded for other complex datasets (apart from CIFAR10)

The rating is primary reflection of above questions and general applicability of this approach for practical use cases. I would consider to change my rating based on the answer to above questions as as well as few points mentioned in weakness section.

---

### Official Review · Reviewer_wULn · 2024-11-01

**Soundness:** 3
**Presentation:** 4
**Contribution:** 2
**Rating:** 8
**Confidence:** 5

**Summary:**

The paper introduces the first attempt at achieving 1-bit fully quantized training (FQT), aka quantizing weights, activation and gradients at 1-bit.

They first provided a theoretical analysis demonstrating that under the assumption of a bounded gradient, the convergence rate for a convex loss is influenced by the variance of the gradient. This informs their hypothesis that the Adam optimizer is better suited for FTQ compared to SGF.

They also show that the quantized gradient variance increases quadratically as the bitwidth decreases, suggesting the variance of 1-bit per-sample gradient quantization is too large for training. To address this,  they propose the average 1-bit per-group quantization that randomly prunes samples proportionally to their dynamic range.  They also propose a similar scheme for weight gradient, called Sample Channel Joint Quantization, that applies the same idea to groups of channels.

They perform various fine-tuning experiments and consistently show they outperform vanilla per-sample quantization and are within an average 5.5% drop for ResNet18 and 5% for VVGNet-16 on a collection of downstream tasks. They also make the first attempt at FTQ from scratch on CIFAR-10/100 and ImageNet, but the results there are way below the QAT. They also test their implementation on Hygon and Raspberry Pi, showing consistent training speedups.

**Strengths:**

A very decent effort by the authors to provide a framework for 1-bit FTQ. To my knowledge, they are the first to undertake that daunting task. Even if the results are disappointing in terms of accuracy, it can be the basis of future work that will further close the gap. I summarise their strengths:

* The first to tackle the difficult task of 1-bit FTQ, even if their contributions are limited IMO.
* Solid presentation and easy-to-read paper.
* A theoretical motivation for their suggested methods and of the optimizers.
* Extensive experimentation covering many tasks, including image classification, object detection and NLP
* A very detailed appendix including ablations studies and implementation details for reproducibility. I specifically like that they consider the generalizability of their approach by applying it to more advanced binary models and showing an improvement even at higher bits.

**Weaknesses:**

Whereas the paper is very well presented and tackles a novel and difficult however, its contribution is quite limited as it mainly builds on theory and methods already developed. Chen 2020 first introduces the relationship between quantized gradient and gradient range found in section 4.2. They motivate their per-sample quantization (PSQ) based on this and introduce the Block Householder Quantizer (BHQ). The authors compare their method to PSQ in their experiments but not to BHQ, which follows the same idea as their average 1-bit quantization.

I also find their conclusion about the convergence properties problematic because there is little evidence to suggest that the loss function of neural networks is convex. In addition, they compare vanilla SGD to Adam without considering momentum SGD first.  Adam does indeed have great properties for training networks, but it’s hardly surprising that SGD without momentum fails in the high variance regime of low-bit quantization. This is why momentum is introduced to act as a low-pass filter.

The authors introduce their FTQ method mainly for on-device fine-tuning. However, they do not discuss certain limitations regarding data movement and memory restrictions on edge devices. According to the appendix, they use large batches of size 64 for on-device fine-tuning (which may not be realistic for real edge applications). In addition, the dequantization operation in the step of algorithm 1 would require storing the large high-precision tensors (assuming this is float32) in memory to first calculate the dynamic ranges and perform the pruning/quantization later.  At such low precision, the training would most likely become memory-bound. It would be useful to see a profiling of training on the device to better understand the impact of the high-precision operations

**Questions:**

* Fig. 1 what is the quantization scheme for the gradient?
* Tables 2 & 3 are very small and hard to read. Can you please expand them a bit?
* It would be interesting to see a comparison of their method to the Per-Sample Quantizer combined with the Block Householder Quantizer for activation gradients.
* A profiling of the training process on either of the edge devices, focusing on the compute time spent on the high-precision operations, e.g. Batch-Norm gradient estimation, and the memory bandwidth of activation gradients.
* Using the symbol of a Hessian $\mathbf{H}$ to denote activations in section 3 is a bit confusing (but this is just a remark)
* The equations in section 3.1 have no numbering. I believe that a quantization function must be applied in line 149 for the equations to hold. The product of two quantize tenses:  $\hat{\nabla}_{\mathbf{H}^{(l)}} \bar{\mathbf{\Theta}}^{(l)^T}$  will not any more be in the same precision as the two inputs. Shouldn't it be instead

 $$Q_g(\hat{\nabla}_{\mathbf{H}^{(l)}} \bar{\mathbf{\Theta}}^{(l)^T})$$

---

### Official Review · Reviewer_VHKA · 2024-11-03

**Soundness:** 3
**Presentation:** 3
**Contribution:** 2
**Rating:** 3
**Confidence:** 5

**Summary:**

This paper explores the limit of fully quantized training (FQT) by proposing a 1-bit quantization scheme for weights, activations, and gradients.

1) The authors introduce an Average 1-bit Quantization (AQ) strategy to reduce gradient variance by pruning less informative gradients.

2) Additionally, they propose Sample Channel joint Quantization (SCQ) to optimize quantization for weight and activation gradients, ensuring compatibility with low-bit hardware.

Experimental results show that the proposed method achieves significant training speedup on VGGNet-16 and ResNet-18 across multiple datasets while maintaining minimal accuracy loss.

**Strengths:**

(1) This paper presents a challenging 1-bit quantization method in the field of FQT, offering valuable insights for future low-bit quantization research through a theoretical analysis of gradient variance.

(2) The proposed approach has been comprehensively evaluated across multiple datasets and models (such as CIFAR-10, CIFAR-100, VGGNet-16, and ResNet-18), effectively demonstrating its validity and generalizability.

(3) The paper is well-structured, with detailed descriptions of the methodology and theoretical foundations, and the experimental results clearly showcase the potential of 1-bit FQT.

**Weaknesses:**

(1) The paper mainly compares traditional full-precision training and the existing PSQ method, lacking direct comparisons with recent low-bit quantization techniques (e.g., 2-bit or 4-bit methods), which limits the generality of its conclusions.

(2) The method’s application mainly focuses on transfer learning for edge devices. Results of training from scratch or for larger models like ResNet-50 etc are weak or missing.

(3) The SCQ method relies on per-channel and per-sample adjustments, which may increase hardware implementation complexity, especially for resource-limited devices. There is a lack of discussion on compatibility and scalability across different hardware.

(4) The proposed method first needs to conduct QAT on ImageNet and then undergoes FQT on downstream datasets for large number of epochs, which constrains the application of the proposed method.

**Questions:**

(1) Can this method be applied to larger models like transformers, or maintain convergence in training-from-scratch scenarios on various models? If not, where do you see the main challenges?

(2) Is the SCQ method scalable on resource-constrained hardware? For deployment of low-power devices, would the implementation complexity lead to additional computational overhead?

(3) What's the batch-size of the FQT on edge devices?

---

### Meta-Review · Area_Chair_eXd5 · 2024-12-23

**Metareview:**

This work proposes fully quantized training (FQT) of NNs using 1-bit precision, provides theoretical convergence analysis of SGD and Adam in this context, and proposes an Average 1-bit quantization strategy for stable training.

The underlying problem undoubtedly is very important and highly challenging and I appreciate the efforts made my the authors. However, this work received mixed reviews (mostly negative) and the primary concerns were

- lack of proper comparison with recent low-bit quantization methods
- weak training from scratch results, evaluation is performed on simple task and small datasets
- limited contribution compared to Chen et al. 2020 (PSQ)
- evaluated only on simple tasks
- lack of other variants of SGD (e.g., momentum) to compare with Adam etc.

I also agree with the concerns above and would like to strongly encourage the authors to incorporate them and submit a revised draft to another top venue as the problem they are working on is of high importance. Unfortunately, at this stage, the work is not polished enough to be published at ICLR.

**Additional Comments On Reviewer Discussion:**

- The main points raised by the reviewers were regarding (1) lack of proper comparison with low-bit methods; (2) weak training from scratch results; (3) limited contribution compared to PSQ; (4) limited practical value; (5) convexity assumption of NN loss function etc.
- The authors did provide several new results during the rebuttal including (1) comparison with Block Householder Qunatizer; (2) included RseNet-50 results; (3) provided new theoretical analysis for the non-convex case etc.
- However, though several of the replies from the authors were appreciated by the reviewers, there was no unanimous agreement towards the acceptance. Primarily because of the limited novelty compared to PSQ and lack of proper comparisons and experimental results on large scale datasets.
- The reviewers appreciated the efforts by the authors during the rebuttal and believe that the new results provided during the rebuttal and the new theoretical justifications will surely add value to the work. However, would also require a careful rewrite and proof-reading.

---

### Decision · Program_Chairs · 2025-01-22

Reject